# POTENTIAL OUTCOMES ESTIMATION UNDER HIDDEN CONFOUNDERS

## ABSTRACT

One of the major challenges in estimating conditional potential outcomes and conditional average treatment effects (CATE) is the presence of hidden confounders. Since testing for hidden confounders cannot be accomplished only with observational data, conditional unconfoundedness is commonly assumed in the literature of CATE estimation. Nevertheless, under this assumption, CATE estimation can be significantly biased due to the effects of unobserved confounders. In this work, we consider the case where in addition to a potentially large observational dataset, a small dataset from a randomized controlled trial (RCT) is available. Notably, we make no assumptions on the existence of any covariate information for the RCT dataset, we only require the outcomes to be observed. We propose a CATE estimation method based on a pseudo-confounder generator and a CATE model that aligns the learned potential outcomes from the observational data with those observed from the RCT. Our method is applicable to many practical scenarios of interest, particularly those where privacy is a concern (e.g., medical applications). Extensive numerical experiments are provided demonstrating the effectiveness of our approach for both synthetic and real-world datasets.

## 1 INTRODUCTION

Estimating treatment effects is of significant interest to various scientific communities, such as in medicine (Glass et al., 2013; Feuerriegel et al., 2024) and social sciences (Imbens & Rubin, 2015; Imbens, 2024) for assessing the efficacy of a policy. Recently, various methods have been developed using machine learning to estimate individual-level treatment effects, also known as the *conditional average treatment effects* (CATE) (Shalit et al., 2017; Alaa & Van Der Schaar, 2017; Wager & Athey, 2018; Shi et al., 2019; Guo et al., 2023; Schweisthal et al., 2024; Fang & Liang, 2024). While these methods have proven successful, their effectiveness in estimating treatment effects can be significantly compromised in real-world applications due to the confounding problem(Kallus et al., 2019; Chor et al., 2024). Confounders are variables that influence both the treatment and the outcome. If not properly controlled for, they can severely bias the potential outcome and treatment effect estimations (Rosenbaum & Rubin, 1983). While it is well-established that treatment effects are identifiable under the assumption of *conditional unconfoundedness* (that is, no hidden confounders), *estimating conditional treatment effects becomes much more challenging under unobserved confounders* (Imbens & Rubin, 2015; Kallus & Zhou, 2018). In some ideal scenarios like Randomized Controlled Trials (RCTs), conditional unconfoundedness might be achieved by design. However, these experiments often require an expensive data collection process. Furthermore, *the conditional unconfoundedness assumption is inherently not falsifiable from observational data alone* (Popper, 2005). For instance, passively collected healthcare databases often lack essential clinical details that can influence treatment decisions made by both doctors and patients, such as subjective evaluations of the severity of a condition or personal lifestyle factors. Consequently, when applying causal inference models to observational data, it is common to assume conditional unconfoundedness, which may fail to hold in practice and cannot be tested. This can cause significant bias in potential outcome estimation.

**Problem Setting.** In this work, we propose a novel approach to mitigate the bias in estimating CATE under hidden confounders. Our analysis begins by considering a scenario in which both observational data and RCT data are present – a common situation in many fields, such as in healthcare, where large observational datasets with rich features (e.g., electronic health records) are readily available, but

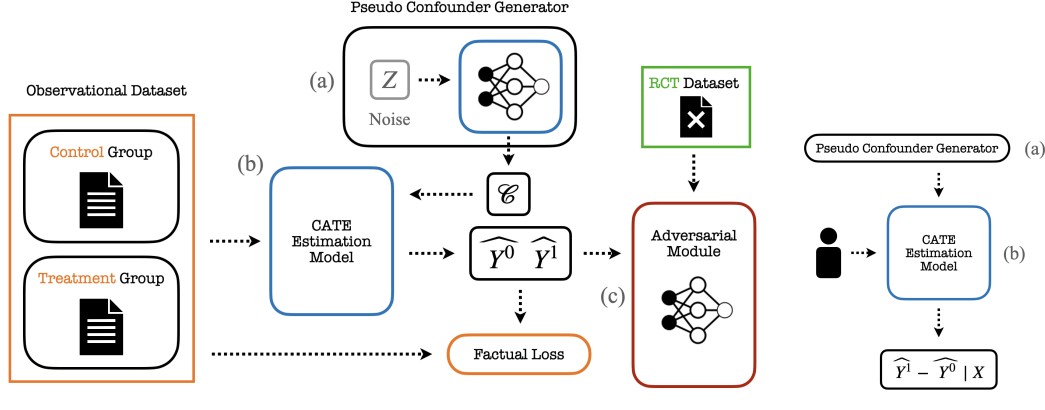

(i) Training procedure.    (ii) Inference procedure.

Figure 1: Schematic of the proposed training and inference procedures. **(i)**: (a) generates pseudo confounders that are used within the CATE estimator using the observational data. Potential outcomes are then matched to the unconfounded RCT dataset in (c). **(ii)**: inference is performed by (a) sampling from the pseudo-confounder generator and (b) using the CATE model with the individual's features.

RCTs are expensive and often too small to support complex models for learning CATE. In particular, we consider scenarios where only the outcomes from a small batch of RCTs are available alongside observational datasets, circumventing the requirements for individual covariates from RCTs. These scenarios include multiple important cases in real-world applications where:

- Full access to the detailed features is unavailable due to privacy concerns;
- Collection of detailed features may be expensive and impractical;
- Requirements of detailed features may introduce selection bias in the RCT design by limiting participation to individuals for whom complete feature information is available.

Therefore, we assume that only the outcomes are accessible in the RCT data.

**Method.** Our proposed method consists of two regularization modules, based on the given outcomes from RCT data, to regularize the search space of hypothesis to prevent bias due to hidden confounders. We note that the proposed regularization modules are CATE model-agnostic, that is, they can be added to any Neural Net-based CATE estimation model.

Marginals Balancing (MB): The first regularization builds on the key fact that the RCT outcomes can be considered as samples from the true potential outcomes. Motivated by this, we use a pseudo-confounder generator to emulate the hidden confounders, based on which the CATE models' predicted potential outcomes should equal in distribution to the observed outcomes from RCT data.

Projections Balancing (PB): The second approach is based on the observation that the projection of the learned potential outcomes onto any transformation of the features should correspond to that of the true potential outcomes on the same transformation.

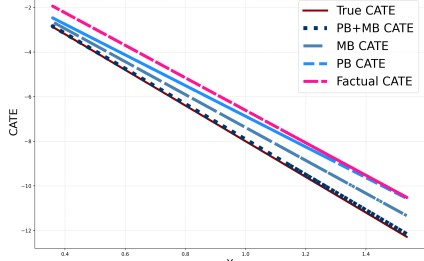

Figure 2: Comparison of CATE estimates using the baseline factual learner, the MB and PB models, and the combined MB+PB model.

Our final model (MB+PB) combines both approaches, as we numerically observe that doing so restricts the search space for the factual optimization problem and achieves the best performance. We illustrate the performance of these different models on a simple Gaussian linear model in Figure 2. See Section 3.1 for a full description of this example.

**Related Works** Several recent works address the challenge of estimating treatment effects under unobserved confounding by combining randomized controlled trials (RCTs) with observational data. Some approaches leverage the internal validity of RCTs and how representative observational data is using techniques such as weighting and doubly robust estimators (Colnet et al., 2024). Other methods propose a linear correction term to adjust for confounding bias (Kallus et al., 2018). Methods have also been developed for estimating heterogeneous treatment effects, requiring covariate-level data for improved accuracy and balancing the representation of different observed features (Hatt et al., 2022a). Kallus et al. (2019) introduce interval estimation for CATE under unobserved confounders and the marginal sensitivity model (Rosenbaum, 2002). It is important to note that all of these methods assume that both individual covariates and outcomes from the RCTs are accessible, which differs from the assumptions of our approach, as we assume that only the outcomes of the RCT are observed. Other methods have explored specific scenarios for estimating CATE from multiple datasets, such as in recommendation systems (Li et al., 2024) or sequential observational data (Hatt & Feuerriegel, 2024). Moreover, recent works have addressed the confounding introduced by applying representation learning approaches to CATE estimation (Melnychuk et al., 2024).

## 2 PROBLEM SETUP

Let $(\Omega, \mathcal{F}, \mathbb{P})$ be a probability space. Consider random variables $(X, U, T, Y_1, Y_0)$ defined on $(\Omega, \mathcal{F}, \mathbb{P})$, where $T$ is a binary random variable denoting treatment assignment, $X \in \mathcal{X} \subset \mathbb{R}^d$ represents the observed features and $U \in \mathcal{U} \subset \mathbb{R}^m$ represents unobserved confounders. The potential outcomes $Y_1, Y_0 \in \mathbb{R}$ correspond to the outcomes under treatment and control, respectively. Let $Y$ represent the observed outcome defined as (Hernán & Robins, 2020)[1]:

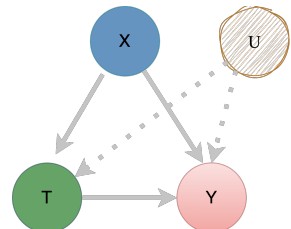

$$Y = TY_1 + (1 - T)Y_0.$$

Figure 3: Causal graph for CATE estimation with unobserved confounders (U).

Figure 3 illustrates the causal graph of these variables.

**Observational Data.** In real scenarios we do not have access to $U, Y_1,$ or $Y_0$ — which gives rise to one of the most fundamental challenges in causal inference. Instead, we only have access to samples of the random triplet $(X, T, Y)$. Thus, we assume an observational dataset $D_o = \{(x_i, t_i, y_i)\}_{i=1}^{n_o}$, consisting of $n_o$ independent observations.

**CATE Estimation.** The objective is to estimate the conditional potential outcomes $\mathbb{E}[Y_t \mid X]$ for $t \in \{0, 1\}$ and CATE $\tau(X)$, defined as:

$$\tau(X) = \mathbb{E}[Y_1 \mid X] - \mathbb{E}[Y_0 \mid X].$$

To this end, we make the standard assumption of *positivity*, that is, $P(T = 1 \mid X) > 0$ almost surely. We also assume that $X \perp\!\!\!\perp U$, which is verified by the causal graph in Figure 3. Moreover, to identify CATE, it is common to assume *conditional unconfoundedness*, that is, $Y_t \perp\!\!\!\perp T \mid X$. While it is well established in the causal inference literature that CATE is identifiable under the assumption of conditional unconfoundedness, this assumption does not hold in the presence of hidden confounders. Without conditional unconfoundedness, CATE is generally not identifiable (Rosenbaum & Rubin, 1983; Imbens & Rubin, 2015). Hidden confounders, which are common in practice, always lead to a violation of the conditional unconfoundedness assumption. Therefore, we focus on scenarios where the conditional unconfoundedness assumption is violated. Specifically, for $t \in \{0, 1\}$, we assume $Y_t \not\!\perp\!\!\!\perp T \mid X$, i.e., the treatment assignment is not independent of the potential outcomes given the observed features due to the presence of unobserved confounders $U$.

**Performance Metric.** Let $\hat{\tau}(x) = h(x, 1) - h(x, 0)$ denote an estimator for CATE where $h$ is a hypothesis $h : \mathcal{X} \times \{0, 1\} \to \mathcal{Y}$ that estimates the conditional potential outcomes $\mathbb{E}[Y_t | X = x]$.

---

[1]Some references take an alternative approach by first defining the factual outcome and then using the consistency assumption to define the potential outcomes.

**Definition 2.1** (PEHE). *The Expected Precision in Estimating Heterogeneous Treatment Effect (PEHE) (Hill, 2011) is defined as:*

$$\varepsilon_{PEHE}(h) = \int_{\mathcal{X}} (\hat{\tau}(x) - \tau(x))^2 p(x) dx \tag{1}$$

*where $p(x)$ is the marginal density of the covariates $X$.*

The $\varepsilon_{\text{PEHE}}$ is widely used as the performance metric for CATE estimation, especially in scenarios where heterogeneous effects are present across different individuals.

**RCT Data.** Given that the bias of hidden confounders cannot even be tested with observational data, we assume access to a small batch of RCT data. In particular, we assume access to only the outcomes of RCT data, instead of the stronger requirement of observing covariates. Let the outcome-only RCT data be denoted as $(T_r, Y_r)$ and let $u = \mathbb{P}(T_r = 1)$. The data generating process of the RCT data is equivalent to the following process: Consider two random variables $Y_1'$ and $Y_0'$ which are equal in distribution to the true potential outcomes $Y_1$ and $Y_0$, respectively. Then with probability $u$, we have one sample of $Y_1'$; with probability $1 - u$, we have one sample of $Y_0'$.

We denote the RCT dataset as $D_r = \{D_r^0, D_r^1\}$ where $D_r^t = \{y_j^t\}_{j=1}^{n_r^t}$ for $t \in \{0, 1\}$. In particular, $D_r^0$ and $D_r^1$ contain $n_r^1$ and $n_r^0$ samples from $Y_1'$ and $Y_0'$.

> The central question we explore in this work is how to apply knowledge about the marginal distributions of the true potential outcomes to help reduce the estimation error of the conditional potential outcomes and CATE under hidden confounders.

We note that to simplify the mathematical analysis we assume that the RCT potential outcomes and the observational data potential outcomes are sampled from the same distribution. However, we will relax this assumption in our empirical setting.

**Confounding Degree.** Additionally, we explore how the *confounding degree*—that is the influence of the unobserved confounder on the treatment assignment—affects the estimation performance. To quantify the degree of unobserved confounding, we employ the commonly used Marginal Sensitivity Model(MSM) (Rosenbaum, 2002). MSM represents a general class of functions that satisfy the $\Gamma$-*selection bias condition* defined as follows.

**Definition 2.2** ($\Gamma$-selection bias condition). *A probability measure $\mathbb{P}$ satisfies the $\Gamma$-selection bias condition with $1 \leq \Gamma < \infty$ if, for $\mathbb{P}$-almost all $u, \tilde{u} \in \mathcal{U}$ and $x \in \mathcal{X}$, the following holds: let $\pi(x, u) = \frac{\mathbb{P}(T=1|x,U=u)}{\mathbb{P}(T=0|x,U=u)}$ and $\pi(x, \tilde{u}) = \frac{\mathbb{P}(T=1|x,U=\tilde{u})}{\mathbb{P}(T=0|x,U=\tilde{u})}$, then*

$$\frac{1}{\Gamma} \leq \frac{\pi(x, u)}{\pi(x, \tilde{u})} \leq \Gamma. \tag{2}$$

The confounding degree is defined as the *minimum value* of $\Gamma$ that satisfies the $\Gamma$-selection bias condition. Specifically, the $\Gamma$-selection condition is satisfied when the odds ratio of receiving the treatment can change by up to a factor of $\Gamma$ as the unobserved confounder $U$ varies, while the observed features remain fixed. Note that when $\Gamma = 1$, this corresponds to the case where $U$ has no effect on the likelihood of treatment assignment given the observed features.

## 3 PROPOSED APPROACH

In this section, we present two models designed to address the challenge of estimating conditional potential outcomes and the CATE in the presence of hidden confounders. To help understand the challenge of hidden confounders, we first discuss in Section 3.1 with a case study about the issue that arises on the baseline factual learner which relies solely on the observational data in the presence of hidden confounders. Next, we introduce our two approaches: Marginals Balancing (MB) in Section 3.2 and Projections Balancing (PB) in Section 3.3. Both approaches are designed to mitigate bias, though they are based on distinct principles. Finally, in Section 3.4, we describe our combined model, MB+PB, which integrates both approaches to improve CATE estimation under hidden confounding.

### 3.1 FACTUAL LEARNER

In the context of conditional potential outcome estimation with observational data, it is standard to solve the following optimization problem based on the observed outcome:

$$\min_{Z_1, Z_0 \ \sigma(X)\text{-measurable}} \mathbb{E}\left[(Z_T - Y)^2\right], \tag{3}$$

where $\sigma(X)$ denotes the $\sigma$-*algebra* generated by $X$. It is well-established ([Theorem 4.1.15] (Durrett, 2019)) that the unique optimal solution (up to a measure zero set) is

$$\forall t \in \{0, 1\}, Z_t^F = \mathbb{E}\left[Y | X, T = t\right],$$

which we will refer to as the factual learner. On the other hand, the goal in causal inference is to learn the conditional potential outcomes $\mathbb{E}\left[Y_t | X\right]$ for $t \in \{0, 1\}$, from which CATE can be computed. Note that under conditional unconfoundedness, we have $Z_t^F = \mathbb{E}\left[Y_t | X\right]$.

However, when conditional unconfoundedness is violated, the solution $Z_t^F$ to the standard optimization problem in Equation (3) does not necessarily equal to $\mathbb{E}\left[Y_t | X\right]$. In other words, the equality $\mathbb{E}\left[Y_t | X\right] = \mathbb{E}\left[Y | X, T = t\right]$ does not necessarily hold. In such cases, the observed data does not provide an accurate estimate of the true treatment effect due to the influence of hidden confounders.

**Case Study.** To empirically illustrate the bias induced by the factual learner, consider the following example. Let the covariate $X$ and the hidden confounder $U$ follow normal distributions where

$$X \sim \mathcal{N}(1.0, 0.04) \quad \text{and} \quad U \sim \mathcal{N}(0, 1).$$

The treatment assignment $T$ is determined by a logistic model that depends on both $X$ and the unobserved confounder $U$:

$$P(T = 1 | X, U) = \frac{1}{1 + \exp(-0.5X - 2U)},$$

The potential outcomes are modeled as linear functions of $X$ and $U$:

$$Y_1 = -3.5X + 3U, \quad Y_0 = 4.5X - 0.6U.$$

The observed outcome $Y$, given by $Y = TY_1 + (1 - T)Y_0$, depends on the treatment assignment $T$.

We sample 1000 samples from $(X, T, Y)$, which is more than sufficient for such a simple problem in a low-dimensional setting, and fit two linear regression models separately on the treatment $(T = 1)$ and control $(T = 0)$ groups, allowing us to estimate the factual learners $\mathbb{E}[Y | X, T = 0]$ and $\mathbb{E}[Y | X, T = 1]$. In Figure 4, we compare the factual learner with the true potential outcomes $\mathbb{E}[Y_t | X]$. This comparison reveals the bias inherent in the factual learner due to the unobserved confounder $U$. In the following sections, we propose two different approaches to alleviate the confounding effect when access to the outcomes of an RCT dataset is available.

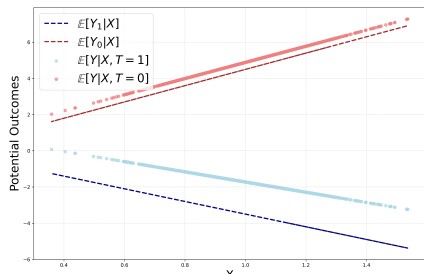

Figure 4: Comparison between the baseline factual learner and the true conditional potential outcomes for a linear Gaussian model.

### 3.2 MARGINALS BALANCING

**Motivation.** To motivate our first model, we begin by observing that the true conditional potential outcomes, $\mathbb{E}[Y_1 | X]$ and $\mathbb{E}[Y_0 | X]$, should ideally correspond to the projection of a random variable sharing the same distribution as the true potential outcomes $Y_1$ and $Y_0$. Specifically, since the true potential outcome $Y_t$ depends on both the covariates $X$ and the hidden confounders $U$, we propose models of the form:

$$\tilde{Y}_t = f_t(X, \tilde{U}),$$

where $f_t : \mathbb{R}^d \times \mathbb{R} \to \mathbb{R}$, and $\tilde{U} \in \mathbb{R}$ is a random variable representing the *pseudo-confounder*. As motivated in Section 2, given the knowledge of the marginal distribution of $Y_t$ (from the RCT outcomes), it is natural to impose the following constraint:

$$\tilde{Y}_t \stackrel{d}{=} Y_t, \tag{4}$$

where $\stackrel{d}{=}$ denotes equality in distribution. Thus, the model $\tilde{Y}_t$ should interpolate the observational data under the constraint in Equation (4).

**Method.** Our first approach, which we refer to as the *Marginals Balancing* (MB), follows this observation and can be formalized through the following optimization problem:

**Definition 3.1** (Optimization Problem of MB). *Let $\mathcal{B}(\mathbb{R})$ denote the set of real-valued continuous and bounded functions. MB solves the following optimization problem:*

$$\min_{Z_1, Z_0 \ \sigma(X)\text{-measurable}} \mathbb{E}\left[(Z_T - Y)^2\right], \tag{5}$$

*where, for $t \in \{0, 1\}$, $Z_t = \mathbb{E}\left[f_t(X, \tilde{U})|X\right]$ for some function $f_t : \mathbb{R}^d \times \mathbb{R} \to \mathbb{R}$ and a random variable $\tilde{U} \in \mathbb{R}$ that conform to the following constraint:*

$$\forall t \in \{0, 1\}, \forall \tilde{g} \in \mathcal{B}(\mathbb{R}), \quad \mathbb{E}\left[\tilde{g}(f_t(X, \tilde{U}))\right] = \mathbb{E}\left[\tilde{g}(Y_t)\right]. \tag{6}$$

Note that the constraint in Equation (6) implies the constraint in Equation (4) due to the Portmanteau Lemma (Billingsley, 1995). It is important to also note that $\mathbb{E}\left[\tilde{g}(Y_t)\right]$ can be estimated with the outcomes in the RCT data because they can be considered as samples of a random variable $Y_t'$ that equal in distribution to $Y_t$.

**Implementation.** To solve the optimization problem of MB, we generate the pseudo-confounder $\tilde{U}$ using a neural network $\psi$, and fit a CATE estimation model $\mu_t(X, \tilde{U})$, with the observed covariates along with the generated pseudo-confounder as inputs, to predict the observed outcomes in the observational dataset $D_o$. Moreover, we enforce that the predicted potential outcomes match the true potential outcomes in distribution. We achieve this by adversarial training, where we instantiate $\mathcal{B}(\mathbb{R})$ with a neural net, and update its parameter to maximize the $L_2$ distance between the right-hand side and the left-hand side of the equality in Equation (6), estimated through the RCT data $D_r$.

**Empirical Illustration.** Figure 5 illustrates the performance of MB model on the case study in Section 3.1. We can observe that the gap between the true conditional potential outcomes and the predicted potential outcomes is indeed reduced compared to the factual learner.

**Limitation.** One notable limitation of the marginal balancing method is that the optimal solution to the MB optimization problem is not unique. Moreover, for certain classes of functions, it is possible to construct an optimal solution under the imposed constraint that does not recover the true conditional potential outcomes, as demonstrated by the example provided in Appendix A.1.

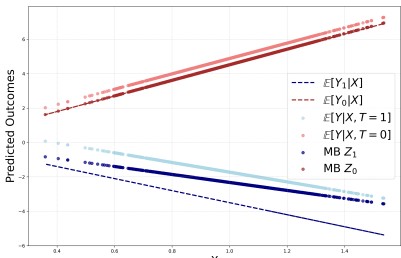

Figure 5: Comparison of the factual learner and MB model with the true conditional potential outcomes.

### 3.3 PROJECTIONS BALANCING

We now introduce our second approach, called *Projections Balancing* (PB).

To illustrate the benefits of this method, we begin by considering an idealized scenario with direct access to the true potential outcomes $Y_1$ and $Y_0$, rather than relying on the RCT data containing samples of $Y_1'$ and $Y_0'$ which are random variables equal in distribution to $Y_1$ and $Y_0$. In practice, this is unattainable since the treatment assignment biases the distribution of the observed outcomes in observational data. We will later relax this learner under the assumption that only a small subset of RCT outcomes is available.

We begin with the following result, which presents a constrained optimization problem whose *unique optimal solution is precisely the conditional potential outcome* $\mathbb{E}[Y_t|X]$, the quantity we aim to identify in causal inference.

**Proposition 3.2** (Ideal PB). *Let* $\mathcal{G} = \{g : \mathbb{R} \to [-1, 1]\}$ *and consider the following optimization problem:*

$$\min_{Z_1, Z_0 \ \sigma(X)\text{-measurable}} \mathbb{E}\left[(Z_T - Y)^2\right]$$

*subject to the constraint*

$$\forall g \in \mathcal{G}, \forall t \in \{0, 1\}, \quad \mathbb{E}[Z_t g(X)] = \mathbb{E}[Y_t g(X)].$$

*The unique solution for this problem is:*

$$\forall t \in \{0, 1\}, \quad Z_t = \mathbb{E}[Y_t \mid X].$$

*Proof of Proposition 3.2.* See in Appendix A.1. $\square$

**Method.** We underscore that the most notable advantage of the ideal PB learner is that *it provides a unique solution corresponding to the true potential outcomes*. Without access to the true potential outcomes in practice, we now introduce a practical PB learner by relaxing the proposed ideal PB learner to scenarios where only RCT outcomes are available.

**Definition 3.3** (Optimization Problem of PB). *Let* $\mathcal{C} \in \mathbb{R}^+$ *be a positive constant and* $\mathcal{G} = \{g : \mathbb{R} \to [-1, 1]\}$. *PB has the following optimizing problem:*

$$\min_{Z_1, Z_0 \ \sigma(X)\text{-measurable}} \mathbb{E}\left[(Z_T - Y)^2\right];$$
$$\text{s.t.} \max_{t \in \{0,1\}} \sup_{g \in \mathcal{G}} \left|\mathbb{E}[Z_t g(X)] - \mathbb{E}[Y_t' g(X)]\right| \leq \mathcal{C}, \tag{7}$$

*where* $Y_t'$ *is a random variable equal in distribution to the true potential outcome* $Y_t$.

In this formulation, the true potential outcomes $Y_t$ are replaced by the RCT potential outcomes $Y_t'$. However, since this problem is challenging to optimize, in practice, we employ the optimization duality and optimize the following optimization problem with a penalty term:

$$\min_{Z_1, Z_0 \ \sigma(X)\text{-measurable}} \left(\mathbb{E}\left[(Z_T - Y)^2\right] + \alpha \sum_{t=0}^{1} \sup_{g \in \mathcal{G}} \left|\mathbb{E}[Z_t g(X)] - \mathbb{E}[Y_t' g(X)]\right|\right) \tag{8}$$

where $\alpha \in \mathbb{R}^+$ is a regularization parameter. We now provide a theoretical guarantee for the PB learner in Equation (7), which characterizes the deviation of the predicted conditional potential outcomes from the true conditional potential outcomes.

**Proposition 3.4** (Practical Projections Balancing (PB)). *Let* $t \in \{0, 1\}$ *and define*

$$L_p(Z_t) = \sup_{g \in \mathcal{G}} \ \left|\mathbb{E}[Z_t g(X)] - \mathbb{E}[Y_t' g(X)]\right|$$

*with* $Y_t' \stackrel{d}{=} Y_t$ *and* $Y_t' \perp\!\!\!\perp Y_t$. *We have that,*

$$\mathbb{E}[|Z_t - \mathbb{E}[Y_t|X]|] \leq L_p(Z_t) + \sqrt{\text{Var}(Y_t)}, \tag{9}$$

*where* $\sqrt{\text{Var}(Y_t)}$ *represents the standard deviation of the potential outcomes.*

*Proof of Proposition 3.4.* See in Appendix A.1. $\square$

**Empirical Illustration.** Figure 6 illustrates the performance of this model on the synthetic linear example in Section 3.1. We can observe that the gap between the true conditional potential outcomes and the predicted potential outcomes is reduced compared to the factual learner.

*Remark* 3.5. In particular, Equation( 9) provides an upper bound on the error of potential outcome estimation of any estimator $Z_t$. It implies that an estimator with low value of $L_p(Z_t)$ is a good estimator of the true conditional potential outcomes. To this end, note that $L_p(Z_t)$ measures how well the estimator $Z_t$ conforms the PB constraint in Equation (7). Thus, a solution to the PB optimization has guaranteed performance. Given that CATE under hidden confounders is not identifiable under general conditions, we conjecture that the standard deviation term in the error bound may not be further reduced due to the *inherent stochasticity of $Y_t$* and the *confounding effects of hidden confounders*.

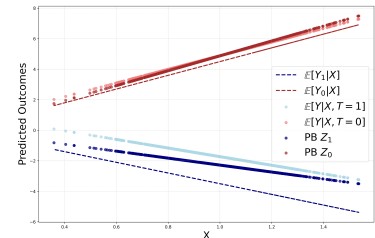

Figure 6: Comparison of the factual learner and PB model with the true conditional potential outcomes.

### 3.4 ALGORITHM: MARGINALS + PROJECTIONS BALANCING

In this section, we present our proposed approach to combine both the Marginals Balancing and Projections Balancing, entitled MB+PB. The rationale behind the effectiveness of our approach is to restrict the search space for the factual optimization objective and to push the solution to get as close as possible to the true conditional potential outcomes.

**Optimization Objective.** The objective function for MB+PB is the following:

$$\min_{Z_1, Z_0 \ \sigma(X)\text{-measurable}} \left( \mathbb{E}\left[ (Z_T - Y)^2 \right] + \alpha \sum_{t=0}^{1} \mathcal{L}_t(f_t) \right),$$

where

$$\mathcal{L}_t\left(f_t\right) = \sup_{g \in \mathcal{G}} \left\| \mathbb{E}\left[ f_t\left(X, \tilde{U}\right) g(X) \right] - \mathbb{E}\left[ Y_t' g(X) \right] \right\| + \sup_{\tilde{g} \in \mathcal{B}} \left\| \mathbb{E}\left[ \tilde{g}(f_t(X, \tilde{U})) \right] - \mathbb{E}\left[ \tilde{g}(Y_t') \right] \right\| \quad (10)$$

and $Z_t = \mathbb{E}\left[ f_t\left(X, \tilde{U}\right) | X \right]$ for some function $f_t$ and a random variable $\tilde{U}$.

**Empirical Illustration.** Figure 7 illustrates the performance of this model on the case study in Section 3.1. We observe that the gap between the true conditional potential outcomes and the predicted potential outcomes is almost entirely reduced. Comparing with the performance of applying MB and PB individually in Figure 5 and 6, MB+PB demonstrates significantly superior performance. Motivated by this, we opt for MB+PB as our final approach.
**Training.** We now present below the general procedure to train the model MB+PB for a general class of functions. For all pseudo-code details, check Algorithm 1.

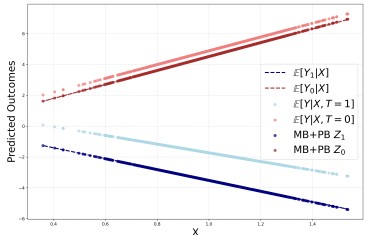

Figure 7: Comparison of the factual learner and MB+PB model with the true conditional potential outcomes.

1. Pseudo-Confounder Generation. We generate Gaussian noise $\eta \in \mathbb{R}^l \sim \mathcal{N}\left(\mathbf{0}, \mathbf{I}\right)$, where $l$ is the dimension of the generated noise. The noise is passed through a neural network generator $\psi$, and we set $\tilde{U} = \psi\left(\eta\right)$.

2. Potential Outcomes Estimation. Both the features $X$ and the generated pseudo-confounder $\tilde{U}$ are fed into a neural network-based conditional potential outcomes learner $f_t$ to have the predicted potential outcome $f_t(X, \tilde{U})$.

3. Balancing. Meanwhile, the predicted potential outcomes $f_1(X, \tilde{U})$ and $f_0(X, \tilde{U})$ are balanced with the RCT outcomes $Y_1'$ and $Y_0'$, respectively, through the regularization defined in Equation (10).

## 4 EMPIRICAL RESULTS

### 4.1 SYNTHETIC EXPERIMENTS

Following Kallus et al. (2019), we begin our empirical evaluation with a synthetic example. This example allows us to control the confounding degree based on a parameter $\Gamma$ of MSM (defined in Section 2.2) and *explore the effect of varying levels of hidden confounding* on the estimation of CATE.

**Data Generating Process.** We consider an one-dimensional example to illustrate the influence of unobserved confounding on estimating CATE. In this example, we generate an unobserved binary confounder $U \sim \text{Bern}(1/2)$, which is independent of other variables, and a covariate $X \sim \text{Unif}[-2, 2]$. The nominal propensity score is defined as $e(x) = \sigma(0.75x + 0.5)$, where $\sigma(\cdot)$ is the logistic sigmoid function. To investigate the impact of confounding, we consider a sensitivity parameter $\Gamma$ and define the complete propensity score as:

$$e(x, u) = u \cdot \alpha_t(x; \Gamma) + (1 - u) \cdot \beta_t(x; \Gamma), \tag{11}$$

with $\alpha_t(x; \Gamma) = \left( \frac{1}{\Gamma \cdot e(x)} \right) + 1 - \frac{1}{\Gamma}$, and, $\beta_t(x; \Gamma) = \left( \frac{\Gamma}{e(x)} \right) + 1 - \Gamma$.

Moreover, the treatment assignment $T$ is sampled as $T \sim \text{Bern}(e(X, U))$. This structure ensures that the complete propensity scores attain the extremal marginal sensitivity model (MSM) bounds corresponding to $\Gamma$ (see (Kallus et al., 2019) for more details). The outcome model is chosen to exhibit a nonlinear CATE, incorporating both linear confounding terms and a noise component $\varepsilon \sim \mathcal{N}(0, 1)$. Specifically, the potential outcome $Y_t$ is defined as:

$$Y_t = (2t - 1)X + 2(2t - 1) - 2\sin(2(2t - 1)X) - 2(2U - 1)(1 + 0.5X) + \varepsilon.$$

**Results.** The results are illustrated in Figure 8. In particular, with increasing confounding level measured by $\log(\Gamma)$, methods such as MB, PB, and the baseline show a marked increase in estimation error. However, MB+PB demonstrates strong robustness and maintains lower errors even at high confounding levels. This suggests that our approach is better equipped to handle the adverse effects of hidden confounders, which is crucial when the confounding degree is unknown. Notably, domain knowledge can only provide very coarse estimations of the confounding degree.

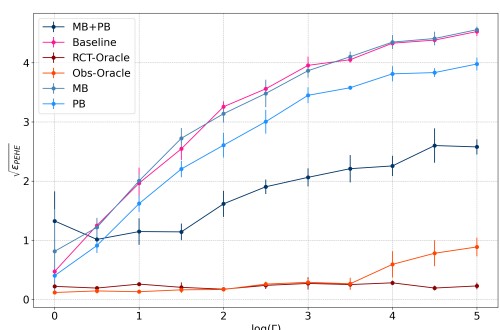

Figure 8: $\sqrt{\varepsilon_{\text{PEHE}}}$ for different confounding degrees. Baseline: Factual Learner, MB: Marginals Balancing, PB: Projections Balancing, MB+PB: Combined Marginals and Projections Balancing, RCT-Oracle: Using a large RCT dataset with covariates, and Obs-Oracle: Using the observational dataset without hidden confounders.

**Influence of RCT Data Size**: In Figure 9, we observe that after using only 50 RCT data points in addition to more than 1000 observational data points, the performance of MB+PB stabilizes. This shows that our model requires only a small number of RCT points to achieve enhanced performance, without requiring the covariates information of RCT data. Even with as few as 25 data points (the sum of both control and treatment units), we can see improved performance over the biased factual learner. It is important to note that this improvement is not observed when RCT points are simply added to the observational data, even when their features are included in training.

### 4.2 REAL DATA APPLICATION

Following the setting of Hatt et al. (2022a), we apply MB+PB to three real-world datasets. We briefly describe them below, with more details deferred to Appendix A.2.1.

**STAR**: A randomized study from 1985 investigating the effect of class size (treatment) on students' standardized test scores (outcome). Following (Kallus et al., 2018), we obtain a dataset with 8 covariates for $4,139$ students: $1,774$ in small classes and $2,365$ in regular classes.

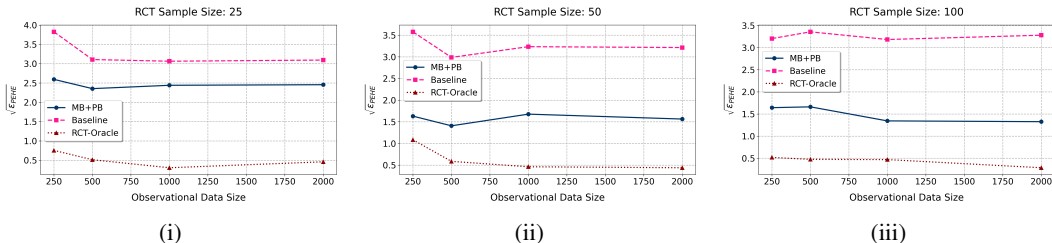

(i)             (ii)             (iii)

Figure 9: Comparison of $\sqrt{\varepsilon_{\text{PEHE}}}$ across different RCT and observational data sample sizes. Baseline: Factual Learner, MB+PB: Combined Marginals and Projections Balancing, and RCT-Oracle. The size of the baseline and RCT-Oracle is equal to the sum of the RCT samples and the observational data size.

**ACTG**: A clinical trial on the effects of different treatments for HIV-1 patients with CD4 counts of 200-500 cells/mm³. The outcome is the change in CD4 counts after $20 \pm 5$ weeks.

**NSW**: An RCT studying the effect of job training on income ((LaLonde, 1986). Following Smith & Todd (2005), we combine 465 randomized subjects (297 treated, 425 control) with 2,490 observational controls, including 8 covariates.

Following the setting in Hatt et al. (2022a), the original dataset is used to estimate pseudo-true potential outcomes, which we treat as the ground truth. Confounding bias is introduced by dropping instances based on outcome thresholds. Further details are in Appendix A.2.2. The RCT data points are sampled from a distributionally different population from the observational population, increasing selection bias. Despite this, our method remains robust.

Table 1: Comparison of $\sqrt{\epsilon_{\text{PEHE}}}$ across three real-world datasets. Results are presented for 10 runs.

|  | $\sqrt{\epsilon_{\text{PEHE}}}$ (Mean $\pm$ Std) | | |
| --- | --- | --- | --- |
| **Estimator** | **STAR** | **ACTG** | **NSW** |
| 2-step ridge | $3.01 \pm 0.01$ | $1.51 \pm 0.01$ | $2.82 \pm 0.02$ |
| 2-step RF | $3.14 \pm 0.03$ | $1.58 \pm 0.07$ | $3.10 \pm 0.12$ |
| 2-step NN | $3.03 \pm 0.02$ | $1.60 \pm 0.02$ | $2.82 \pm 0.02$ |
| Baseline | $2.66 \pm 0.01$ | $1.08 \pm 0.04$ | $0.85 \pm 0.04$ |
| CorNet | $0.59 \pm 0.01$ | $0.42 \pm 0.06$ | $0.14 \pm 0.07$ |
| CorNet$^+$ | $0.38 \pm 0.07$ | $\mathbf{0.27} \pm 0.03$ | $0.21 \pm 0.08$ |
| MB+PB (Ours) | $\mathbf{0.36} \pm 0.04$ | $0.52 \pm 0.05$ | $\mathbf{0.08} \pm 0.02$ |

**Results.** To assess the effectiveness of our approach in utilizing RCT data, we compare it with the factual learner (*Baseline*) which trains only on observational data, and with methods that use covariate information from RCT data, including *2-step ridge*, *2-step RF*, and *2-step NN* from Kallus et al. (2018), and CorNet models (*CorNet* and *CorNet+*), developed by Hatt et al. (2022a). Table 1 shows that models such as *2-step ridge*, *2-step RF*, and *2-step NN* underperform due to the high variance introduced by inverse propensity score re-weighting, as noted in Hatt et al. (2022a). The CorNet models perform significantly better and are comparable to our approach MB+PB. We emphasize that our MB+PB model relies solely on RCT data outcomes yet still achieves competitive results, outperforming CorNet in two of the three total tasks.

## 5 CONCLUSION

In this work, we introduced two approaches, Marginals Balancing (MB) and Projections Balancing (PB), to address the challenge of CATE estimation under hidden confounders. By leveraging outcome-only RCT data, we demonstrated how these models mitigate bias from unobserved confounders, outperforming benchmark methods. The combination of MB and PB (MB+PB) leads to further enhanced performance across synthetic and real-world datasets. While our methods show promising empirical results, we aim to pursue a deeper theoretical understanding of the proposed methods in future works.

**Ethics Statement.** This work focuses on improving the design of machine learning models for estimating treatment effects. We do not foresee any immediate ethical concerns.

**Reproducibility Statement.** We have provided detailed information on how the datasets are processed and how the models are trained, including hyperparameters values. Additionally, we have included the implementation of our algorithms in Python in the supplementary material.

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

# A APPENDIX

## A.1 PROOFS OF THEORETICAL RESULTS

We begin by presenting an example demonstrating that the optimal solution for the Marginals Balancing objective is not necessarily the true conditional potential outcomes. We then proceed to provide propositions that support the use of the Projections Balancing method.

**Example** Consider the random variables $T, X, Y_0, Y_1$, where $T$ is a binary treatment indicator, $X \in \mathcal{X}$, and $Y_0, Y_1$ are the potential outcomes. We aim to minimize the following MB objective:

$$\mathbb{E}\left[(1-T)\left(\mathbb{E}[\tilde{Y}_0 \mid X] - Y_0\right)^2 + T\left(\mathbb{E}[\tilde{Y}_1 \mid X] - Y_1\right)^2\right],$$

subject to the constraint that $\tilde{Y}_0 \stackrel{d}{=} Y_0$ and $\tilde{Y}_1 \stackrel{d}{=} Y_1$.

Suppose $X \sim \text{Ber}(1/2)$ and $T \sim \text{Ber}(1/2)$, with $T$ and $X$ being independent. Define the potential outcomes as:

$$Y_0 = Y_1 = (1-T)X + T(1-X).$$

Now, consider the random variables $\tilde{Y}_0 = X$ and $\tilde{Y}_1 = 1 - X$. We observe that both $\tilde{Y}_0$ and $\tilde{Y}_1$ satisfy the equality in distribution constraint: $\tilde{Y}_0 \stackrel{d}{=} Y_0$ and $\tilde{Y}_1 \stackrel{d}{=} Y_1$.

Furthermore, we have:

$$\mathbb{E}[\tilde{Y}_0 \mid X](1-T) = X(1-T) = Y_0(1-T),$$

and

$$\mathbb{E}[\tilde{Y}_1 \mid X]T = (1-X)T = Y_1 T.$$

Therefore, the MB objective is minimized, and the objective value is zero. While we have that for the true conditional potential outcomes $\mathbb{E}[Y_1|X]$ and $\mathbb{E}[Y_0|X]$, we have that:

$$\mathbb{E}[Y_1|X] = \mathbb{E}[(1-T)X \mid X] + \mathbb{E}[T(1-X) \mid]$$
$$= \mathbb{E}[1-T]\mathbb{E}[X \mid X] + \mathbb{E}[T]\mathbb{E}[(1-X) \mid X]$$
$$= \frac{1}{2}X + \frac{1}{2}(1-X)$$

Therefore,

$$\mathbb{E}[Y_1|X] = \frac{1}{2}, \qquad \mathbb{E}[Y_0|X] = \frac{1}{2}$$

Which does not achieve a zero loss for the objective.

***Proposition 3.2*** (Ideal Potential outcomes learner 2). Let $(\Omega, \mathcal{F}, \mathbb{P})$ be a probability space. Consider the real random variables $(X, U, T, Y_0, Y_1)$, where $T$ is a binary random variable, and $Y_1, Y_0 \perp\!\!\!\perp T \mid (X, U)$, $Y$ is defined as $Y = TY_1 + (1-T)Y_0$. We also assume that $X \perp\!\!\!\perp U$. We aim to solve the following optimization problem:

$$\min_{Z_1, Z_0 \ \sigma(X)\text{-measurable}} \mathbb{E}\left[(Z_T - Y)^2\right]$$

subject to the constraint

$$\forall g : \mathbb{R} \to [-1, 1], \forall t \in \{0, 1\}, \quad \mathbb{E}[Z_t g(X)] = \mathbb{E}[Y_t g(X)].$$

The unique solution for this problem is

$$\forall t \in \{0, 1\}, \quad Z_t = \mathbb{E}[Y_t \mid X].$$

*Proof of Proposition 3.2.*
We begin with the following identities for the observed and predicted outcomes:

$$Y = TY_1 + (1-T)Y_0, \quad Z_T = TZ_1 + (1-T)Z_0.$$

Thus, the objective function can be expanded as:

$$\mathbb{E}\left[(Z_T - Y)^2\right] = \mathbb{E}\left[(T(Z_1 - Y_1) + (1 - T)(Z_0 - Y_0))^2\right]$$
$$= \mathbb{E}\left[T(Z_1 - Y_1)^2 + (1 - T)(Z_0 - Y_0)^2\right]$$
$$+ 2\mathbb{E}\left[T(1 - T)(Z_1 - Y_1)(Z_0 - Y_0)\right].$$

Since $T \in \{0, 1\}$, we have $T(1 - T) = 0$, so the cross term vanishes:

$$\mathbb{E}\left[T(1 - T)(Z_1 - Y_1)(Z_0 - Y_0)\right] = 0.$$

Thus, the objective simplifies to:

$$\mathbb{E}\left[(Z_T - Y)^2\right] = \mathbb{E}\left[T(Z_1 - Y_1)^2\right] + \mathbb{E}\left[(1 - T)(Z_0 - Y_0)^2\right].$$

Next, we can analyze the optimization for $Z_1$ and $Z_0$ separately. Without loss of generality, we first focus on $Z_1$.

We expand the term for $Z_1$:

$$\mathbb{E}[T(Z_1 - Y_1)^2] = \mathbb{E}[T(Z_1 - \mathbb{E}[Y_1 \mid X] + \mathbb{E}[Y_1 \mid X] - Y_1)^2]$$
$$= \underbrace{\mathbb{E}\left[T(Z_1 - \mathbb{E}[Y_1 \mid X])^2\right]}_{\text{Minimized at zero when } Z_1 = \mathbb{E}[Y_1|X]} + \underbrace{\mathbb{E}\left[T(\mathbb{E}[Y_1 \mid X] - Y_1)^2\right]}_{\text{Independent of the optimization objective}}$$
$$+ \underbrace{2\mathbb{E}\left[T(Z_1 - \mathbb{E}[Y_1 \mid X])(\mathbb{E}[Y_1 \mid X] - Y_1)\right]}_{\text{We prove this term is zero below}}$$

Since $Y_1 \perp\!\!\!\perp T \mid (X, U)$, we have:

$$\mathbb{E}\left[T(Z_1 - \mathbb{E}[Y_1 \mid X])(\mathbb{E}[Y_1 \mid X] - Y_1)\right] = \mathbb{E}\left[(Z_1 - \mathbb{E}[Y_1 \mid X])\pi(X, U)\Psi(U)\right],$$

where $\pi(X, U) = \mathbb{E}[T \mid X, U] \in (0, 1)$ and $\Psi(U) = -\mathbb{E}[Y_1 \mid U]$. Let $A = \{\omega \mid Z_1 - \mathbb{E}[Y_1 \mid X] > 0\}$ and $B = \{\omega \mid \Psi(U) > 0\}$.

We decompose the expectation as follows:

$$\mathbb{E}\left[\pi(X, U)\Psi(U)(Z_1 - \mathbb{E}[Y_1 \mid X])\right] = \mathbb{E}\left[\pi(X, U)\Psi(U)\mathbb{1}_{A \cap B}(Z_1 - \mathbb{E}[Y_1 \mid X])\right]$$
$$+ \mathbb{E}\left[\pi(X, U)\Psi(U)\mathbb{1}_{A^C \cap B}(Z_1 - \mathbb{E}[Y_1 \mid X])\right]$$
$$+ \mathbb{E}\left[\pi(X, U)\Psi(U)\mathbb{1}_{A \cap B^C}(Z_1 - \mathbb{E}[Y_1 \mid X])\right]$$
$$+ \mathbb{E}\left[\pi(X, U)\Psi(U)\mathbb{1}_{A^C \cap B^C}(Z_1 - \mathbb{E}[Y_1 \mid X])\right]$$

We now handle each of these four terms separately:

Case 1 $(A \cap B)$:

This term is positive, as both $Z_1 - \mathbb{E}[Y_1 \mid X] > 0$ and $\Psi(U) > 0$, and since $X \perp\!\!\!\perp U$, we have that:

$$0 \leq \mathbb{E}\left[\pi(X, U)\Psi(U)\mathbb{1}_{A \cap B}(Z_1 - \mathbb{E}[Y_1 \mid X])\right] \leq \mathbb{E}\left[\Psi(U)\mathbb{1}_{A \cap B}(Z_1 - \mathbb{E}[Y_1 \mid X])\right].$$
$$\leq \mathbb{E}\left[\Psi(U)\mathbb{1}_B\right]\mathbb{E}\left[(Z_1 - \mathbb{E}[Y_1 \mid X])\mathbb{1}_A\right]$$
$$\leq \mathbb{E}\left[\Psi(U)\mathbb{1}_B\right]\left(\mathbb{E}\left[Z_1\mathbb{1}_A\right] - \mathbb{E}\left[\mathbb{E}[Y_1\mathbb{1}_A \mid X]\right]\right)$$
$$\leq \mathbb{E}\left[\Psi(U)\mathbb{1}_B\right]\left(\mathbb{E}\left[Z_1\mathbb{1}_A\right] - \mathbb{E}[Y_1\mathbb{1}_A]\right)$$

However, since $\mathbb{1}_A$ is $\sigma(X)$-measurable, we can write it as a function of $X$, more precisely we can choose $g$ to be, $g_A(X) = \mathbb{1}(X \in A)$, therefore,

$$0 \leq \mathbb{E}\left[\pi(X, U)\Psi(U)\mathbb{1}_{A \cap B}(Z_1 - \mathbb{E}[Y_1 \mid X])\right]$$
$$\leq \mathbb{E}\left[\Psi(U)\mathbb{1}_B\right]\left(\mathbb{E}\left[Z_1 g_A(X)\right] - \mathbb{E}[Y_1 g_A(X)]\right) = 0$$

Case 2 $\left(A^C \cap B\right)$:

In this case, $Z_1 - \mathbb{E}[Y_1 \mid X] \leq 0$ and $\Psi(U) > 0$, making this term non-positive:

$$0 \geq \mathbb{E}\left[\pi(X, U)\Psi(U)\mathbb{1}_{A^C \cap B}(Z_1 - \mathbb{E}[Y_1 \mid X])\right] \geq \mathbb{E}\left[\Psi(U)\mathbb{1}_{A^C \cap B}(Z_1 - \mathbb{E}[Y_1 \mid X])\right].$$

Again, by the same reasoning as in Case 1, we have:

$$\mathbb{E}\left[\mathbb{E}\left[(Z_1\mathbb{1}_{A^C} - Y_1\mathbb{1}_{A^C}) \mid X\right]\right] = 0,$$

so this term is also zero.

Case 3 $\left(A \cap B^C\right)$:

Here, $Z_1 - \mathbb{E}[Y_1 \mid X] > 0$ but $\Psi(U) \leq 0$, so this term is non-positive:

$$0 \geq \mathbb{E}\left[\pi(X, U)\Psi(U)\mathbb{1}_{A \cap B^C}(Z_1 - \mathbb{E}[Y_1 \mid X])\right] \geq \mathbb{E}\left[\Psi(U)\mathbb{1}_{A \cap B^C}(Z_1 - \mathbb{E}[Y_1 \mid X])\right].$$

As in the previous cases, we factor out $\mathbb{E}\left[Z_1\mathbb{1}_A - Y_1\mathbb{1}_A \mid X\right] = 0$, so this term is zero.

Case 4 $\left(A^C \cap B^C\right)$:

Finally, in this case, both $Z_1 - \mathbb{E}[Y_1 \mid X] \leq 0$ and $\Psi(U) \leq 0$, so the term is positive:

$$0 \leq \mathbb{E}\left[\pi(X, U)\Psi(U)\mathbb{1}_{A^C \cap B^C}(Z_1 - \mathbb{E}[Y_1 \mid X])\right] \leq \mathbb{E}\left[\Psi(U)\mathbb{1}_{A^C \cap B^C}(Z_1 - \mathbb{E}[Y_1 \mid X])\right].$$

Once again, we apply the same reasoning, and the term equals zero:

$$\mathbb{E}\left[\mathbb{E}\left[(Z_1\mathbb{1}_{A^C} - Y_1\mathbb{1}_{A^C}) \mid X\right]\right] = 0.$$

Thus, each of the four terms is equal to zero. Therefore, the entire expression simplifies to zero:

$$2\mathbb{E}\left[T(Z_1 - \mathbb{E}[Y_1 \mid X])(\mathbb{E}[Y_1 \mid X] - Y_1)\right] = 0.$$

A symmetric argument holds for $Z_0$. By expanding $\mathbb{E}\left[(1 - T)(Z_0 - Y_0)^2\right]$, we can use the same reasoning to show that $Z_0 = \mathbb{E}[Y_0 \mid X]$ minimizes the objective function.

We now observe that $\mathbb{E}\left[(1 - T)(Z_0 - Y_0)^2\right]$, and $Z_0 = \mathbb{E}[Y_0 \mid X]$ verify the constraint as we have for every $g \in \mathcal{G}$:

$$\mathbb{E}\left[\mathbb{E}\left[Y_t \mid X\right] g(X)\right] = \mathbb{E}\left[\mathbb{E}\left[Y_t g(X) \mid X\right]\right]$$
$$= \mathbb{E}\left[Y_t g(X)\right]$$

Combining these results, we conclude the minimizer of the objective function must satisfy:

$$Z_1 = \mathbb{E}[Y_1 \mid X] \quad \text{and} \quad Z_0 = \mathbb{E}[Y_0 \mid X].$$

$\square$

**Proposition 3.4** (Relaxed potential outcomes learner (PB)). Let $\mathcal{G} = \{g : \mathbb{R}^d \to [-1, 1]\}$ and let,

$$L_p(Z_t) = \sup_{g \in \mathcal{G}} |\mathbb{E}\left[Z_t g(X)\right] - \mathbb{E}\left[Y'_t g(X)\right]|$$

with $Y'_t \stackrel{d}{=} Y_t$ and $Y'_t \perp\!\!\!\perp Y_t$. Then,

$$\mathbb{E}\left[|Z_t - \mathbb{E}[Y_t \mid X]|\right] \leq L_p(Z_t) + \sqrt{Var(Y_t)}.$$

*Proof of Proposition 3.4.*
First define

$$L_I(Z_t) = \sup_{g \in \mathcal{G}} |\mathbb{E}\left[Z_t g(X)\right] - \mathbb{E}\left[Y_t g(X)\right]|.$$

We will first prove that

$$\mathbb{E}\left[|Z_t - \mathbb{E}[Y_t \mid X]|\right] \leq L_I(Z_t).$$

Since $Z_t - \mathbb{E}[Y_t \mid X]$ is $\sigma(X)$-measurable, let $A = \{\omega \in \Omega \mid Z_t - \mathbb{E}[Y_t \mid X] > 0\}$ and $B = \{\omega \in \Omega \mid Z_t - \mathbb{E}[Y_t \mid X] \leq 0\}$. We can then define a function $\tilde{g} \in \mathcal{G}$ such that $\tilde{g} = \mathbb{1}_A - \mathbb{1}_B$. We have:

$$|\mathbb{E}\left[Z_t\tilde{g}(X)\right] - \mathbb{E}\left[Y_t\tilde{g}(X)\right]| = |\mathbb{E}\left[(Z_t - Y_t)\tilde{g}(X)\right]|$$
$$= |\mathbb{E}\left[\mathbb{E}\left[(Z_t - Y_t)\tilde{g}(X) \mid X\right]\right]|$$
$$= |\mathbb{E}\left[\mathbb{E}\left[(Z_t - Y_t) \mid X\right]\tilde{g}(X)\right]|$$
$$= \mathbb{E}\left[|\mathbb{E}\left[Z_t - Y_t \mid X\right]\mathbb{1}_A|\right] + \mathbb{E}\left[|\mathbb{E}\left[Z_t - Y_t \mid X\right]\mathbb{1}_B|\right] \quad (A \cup B = \Omega)$$
$$= \mathbb{E}\left[|Z_t - \mathbb{E}[Y_t \mid X]|\right].$$

Since we have

$$|\mathbb{E}\left[Z_t \tilde{g}(X)\right] - \mathbb{E}\left[Y_t \tilde{g}(X)\right]| \le \sup_{g \in \mathcal{G}} |\mathbb{E}\left[Z_t g(X)\right] - \mathbb{E}\left[Y_t g(X)\right]|,$$

it follows that

$$\mathbb{E}\left[|Z_t - \mathbb{E}[Y_t \mid X]|\right] \le L_I(Z_t).$$

Next, we observe:

$$
\begin{aligned}
L_I(Z_t) &= \sup_{g \in \mathcal{G}} |\mathbb{E}\left[Z_t g(X)\right] - \mathbb{E}\left[Y'_t g(X)\right] + \mathbb{E}\left[Y'_t g(X)\right] - \mathbb{E}\left[Y_t g(X)\right]| \\
&\le \sup_{g \in \mathcal{G}} |\mathbb{E}\left[Z_t g(X)\right] - \mathbb{E}\left[Y'_t g(X)\right]| + \sup_{g \in \mathcal{G}} |\mathbb{E}\left[Y'_t g(X)\right] - \mathbb{E}\left[Y_t g(X)\right]| \\
&\le L_p(Z_t) + \sup_{g \in \mathcal{G}} |\mathbb{E}\left[Y'_t\right] \mathbb{E}\left[g(X)\right] - \mathbb{E}\left[Y_t g(X)\right]| \\
&= L_p(Z_t) + \sup_{g \in \mathcal{G}} |\mathbb{E}[Y_t]\mathbb{E}[g(X)] - \mathbb{E}[Y_t g(X)]| \\
&= L_p(Z_t) + \sup_{g \in \mathcal{G}} |\mathrm{Cov}(Y_t, g(X))| \\
&\le L_p(Z_t) + \sqrt{\mathrm{Var}(Y_t)} \sup_{g \in \mathcal{G}} \sqrt{\mathrm{Var}(g(X))} \quad \text{(Cauchy-Schwarz)} \\
&\le L_p(Z_t) + \sqrt{\mathrm{Var}(Y_t)} \quad \text{(Popoviciu's inequality)}
\end{aligned}
$$

Thus, we conclude:

$$\mathbb{E}\left[|Z_t - \mathbb{E}[Y_t \mid X]|\right] \le L_p(Z_t) + \sqrt{Var(Y_t)}.$$

$\square$

## A.2 DATASETS DESCRIPTION

### A.2.1 THE ORIGINAL DATASETS

**Tennessee Student/Teacher Achievement Ratio (STAR) Experiment**  This experiment, initiated in 1985, was designed as a randomized trial to investigate the impact of class size (i.e., the treatment) on students' standardized test performance (i.e., the outcome). At the beginning of the study, students and teachers were randomly allocated to different class sizes, with efforts to maintain these class sizes throughout the experiment. This dataset has been used previously by Kallus et al. (2018) to address bias from unmeasured confounding in observational studies.

In line with Kallus et al. (2018), we focus on two treatment conditions: small classes (13-17 students) and regular-sized classes (22-25 students). The treatment variable is the class size to which students were assigned in the first grade, comprising a total of $4,509$ students. The outcome variable $Y$ is measured as the aggregate score from listening, reading, and mathematics standardized tests administered at the end of the first grade. In addition to class size and test scores, the dataset includes several covariates for each student: gender, race, birth month, birth date, birth year, eligibility for free lunch, rural/urban status, and teacher identification number. After excluding students with incomplete data, the resulting sample consists of $4,139$ students, with $1,774$ assigned to the treatment group (small classes, $T = 1$) and $2,365$ to the control group (regular classes, $T = 0$). We sample

**AIDS Clinical Trial Group (ACTG) Study 175**  The AIDS Clinical Trial Group (ACTG) Study 175 was a randomized clinical trial conducted to compare four treatment regimens on $2,139$ HIV-1-infected patients with CD4 counts between 200 and 500 cells/mm$^3$ (Hammer et al., 1996). The trial compared the effectiveness of zidovudine (ZDV) monotherapy, didanosine (ddI) monotherapy, ZDV combined with ddI, and ZDV combined with zalcitabine (ZAL). This dataset was also used in Hatt et al. (2022b) to study the problem of learning policies that generalize to target populations, making it a challenging candidate for evaluating our method due to underrepresentation of certain subgroups, such as HIV-positive females, in clinical trials (Gandhi et al., 2005; Greenblatt, 2011).

The outcome $Y$ in this dataset is defined as the change in CD4 count from the start of the study to $20 \pm 5$ weeks later. The estimated average treatment effects for male and female subgroups are $-8.97$ and $-1.39$, respectively (Hatt et al., 2022b), indicating a notable difference in treatment response

between genders. We focus on two treatment arms: the combined ZDV and ZAL treatment ($T = 1$) and ZDV monotherapy ($T = 0$). The dataset comprises $1,056$ patients with 12 covariates, including five continuous variables: age (years), weight (kg, denoted as wtkg), baseline CD4 count (cells/mm$^3$), Karnofsky score ($0 - 100$ scale, denoted as karnof), and baseline CD8 count (cells/mm$^3$). All continuous variables are centered and scaled prior to analysis. The dataset also includes seven binary covariates: gender ($1 =$ male, $0 =$ female), homosexual activity (homo, $1 =$ yes, $0 =$ no), race ($1 =$ nonwhite, $0 =$ white), intravenous drug use history (drug, $1 =$ yes, $0 =$ no), symptomatic status (symptom, $1 =$ symptomatic, $0 =$ asymptomatic), antiretroviral experience (str2, $1 =$ experienced, $0 =$ naive), and hemophilia (hemo, $1 =$ yes, $0 =$ no).

**National Supported Work (NSW) Demonstration** The National Supported Work (NSW) Demonstration was a subsidized work program that ran for four years across 15 locations in the United States, providing participants with transitional work experience and assistance in securing regular employment. From April 1975 to August 1977, the NSW program operated as a randomized experiment in 10 locations, with some applicants randomly assigned to a control group that did not participate in the program. Data for $6,616$ treatment and control observations were collected through retrospective baseline interviews and four follow-up interviews, covering a two-year period before randomization and up to 36 months afterward.

For our analysis, we use a randomized dataset from LaLonde (1986), following the setup of Smith & Todd (2005). We combine randomized samples from 465 subjects (297 treated and 425 controls) with 2,490 control samples from the Panel Study of Income Dynamics (PSID) to create an observational dataset. The resulting dataset consists of 297 treated observations ($T = 1$) and 2,915 control observations ($T = 0$). This study includes 8 covariates: age, education level, ethnicity (represented as two variables), marital status, and educational attainment.

### A.2.2 GENERATING SMALL RANDOMIZED OUTCOMES AND LARGE OBSERVATIONAL DATASETS

In line with the method used by Kallus et al. (2018); Hatt et al. (2022a) we generate a large observational dataset with confounding and a smaller unconfounded randomized dataset consisting solely of the outcomes, both derived from the real-world data described in Section A.2.1. Importantly, the randomized dataset is drawn from a different population than the observational one, reflecting the limitations of randomized controlled trials (RCTs) in generalizing to the broader population of interest.

To do this, we follow the same procedure for the STAR, ACTG, and NSW datasets. First, we generate a small, unconfounded randomized dataset by sampling a small fraction of the RCT data points $128, 50, 50$. instances from the original dataset. We introduce a distributional discrepancy between the randomized and observational datasets by selecting individuals for the randomized dataset based on a covariate ("birthday" for STAR, "gender" for ACTG, and "age" for NSW), see (Hatt et al., 2022a) for further details. Second, we create the observational dataset by introducing unobserved confounding, ensuring that the treatment and control groups differ systematically in their potential outcomes. Following Kallus et al. (2018), we select subjects from those who were not included in the randomized dataset: controls ($T = 0$) with especially low outcomes (i.e., $y_i < \mathbb{E}[Y \mid T = 0] - c \cdot \sigma_{Y|T=0}$, where $\sigma_{Y|T=0}$ is the standard deviation of the outcomes in the control group) and treated subjects ($T = 1$) with notably high outcomes (i.e., $y_i > \mathbb{E}[Y \mid T = 1] + c \cdot \sigma_{Y|T=1}$, where $\sigma_{Y|T=1}$ is the standard deviation of the outcomes in the treatment group).

The constant $c$ is adjusted according to the size of the original dataset (with $c = 1$ for STAR, $c = 0$ for ACTG, and $c = 0.25$ for NSW) to control the number of subjects in the observational dataset, ensuring that it remains large. This process introduces confounding by selectively including control subjects with lower outcomes and treated subjects with higher outcomes into the observational treatment and control groups. As a result, a naïve estimator relying solely on the observational data will be biased. Moreover, because this selection is based on the outcome variable, it becomes impossible to control for this confounding.

---

**Algorithm 1** Training Algorithm for Marginals and Projections Balancing (MB+PB)

---

1: **Input:** $D_o = \{(x_i, t_i, y_i)\}_{i=1}^{n_o}$, $D_r = \{D_r^0, D_r^1\}$ where $D_r^t = \{y_j^t\}_{j=1}^{n_r^t}$ for $t \in \{0, 1\}$, initial and final weights $(\alpha_s, \alpha_e)$, number of epochs $N_2$, balancing iterations $N_b$, neural networks for: potential outcomes $(\mu)$, marginals balancing $(\tilde{g})$, and projections balancing $(g)$.
2: **Output:** Trained models $\mu$ and $\psi$.
3: Initialize noise $\eta \sim \mathcal{N}(\mathbf{0}_l, \mathbf{I}_l)$ and generate $n_o$ samples $\{\eta_i\}_{i=1}^{n_o}$.
4: **for** epoch $= 1$ to $N_1$ **do**
5:     Increase $\alpha$ from $\alpha_s$ to $\alpha_e$.
6:     Generate noise $\tilde{u}_i = \psi(\eta_i)$ and estimate outcomes $\hat{y}_i = \mu_{t_i}(x_i, \tilde{u}_i)$ for all $1 \le i \le n_o$.
7:     Compute factual loss:

$$\mathcal{L}_f = \frac{1}{n_o} \sum_{i=1}^{n_o} \left( t_i (y_i - \hat{y}_i)^2 + (1 - t_i)(y_i - \hat{y}_i)^2 \right)$$

8:     Generate potential outcomes $\hat{y}_i^1 = \mu_1(x_i, \tilde{u}_i)$ and $\hat{y}_i^0 = \mu_0(x_i, \tilde{u}_i)$.
9:     Compute marginals balancing loss:

$$\mathcal{L}_m = \left( \frac{1}{n_r^1} \sum_{i=1}^{n_r^1} \tilde{g}(y_i^1) - \frac{1}{n_o} \sum_{i=1}^{n_o} \tilde{g}(\hat{y}_i^1) \right)^2 + \left( \frac{1}{n_r^0} \sum_{i=1}^{n_r^0} \tilde{g}(y_i^0) - \frac{1}{n_o} \sum_{i=1}^{n_o} \tilde{g}(\hat{y}_i^0) \right)^2$$

10:     Compute projections balancing loss:

$$\mathcal{L}_p = \left( \frac{1}{n_r^1} \sum_{i=1}^{n_r^1} g(x_{\lambda(i)}) y_i^1 - \frac{1}{n_o} \sum_{i=1}^{n_o} g(x_i) \hat{y}_i^1 \right)^2 + \left( \frac{1}{n_r^0} \sum_{i=1}^{n_r^0} g(x_{\lambda(i)}) y_i^0 - \frac{1}{n_o} \sum_{i=1}^{n_o} g(x_i) \hat{y}_i^0 \right)^2$$

    where $\lambda(i)$ selects a random number between 1 and $n_o$.
11:     Compute total loss $\mathcal{L} = \mathcal{L}_f + \alpha(\mathcal{L}_m + \mathcal{L}_p)$
12:     Backpropagate to update $\mu$ and $\psi$ using Adam.
13:     **for** each balancing iteration $n = 1$ to $N_{\text{balancing}}$ **do**
14:         Calculate the negative regularization loss: $\mathcal{L}_r = -(\mathcal{L}_m + \mathcal{L}_p)$
15:         Backpropagate to update $\tilde{g}$ and $g$ using Adam.
16:     **end for**
17: **end for**
18: Return trained models $\{\mu_t\}_{t=0}^1$, and $\psi$.

---

### A.3    IMPLEMENTATION DETAILS

In this section, we provide the implementation details of our proposed algorithm MB+PB. Specifically, we describe the neural network architectures used for the different modules in our algorithm. Additionally, we present a detailed pseudo-code for the training procedure.

**The Neural Networks Architectures.**    As detailed in Section 3.4, MB+PB consists of three components: a generator $\psi(\eta)$, a CATE learner $\mu_t(X, \tilde{U})$, a marginals balancing module $\tilde{g}$, and a projections balancing module $g$.

- **Pseudo-Confounder Generator:** The generator $\psi(\eta)$ is a neural network designed to generate pseudo-confounders from the input variables, which consist of standard Gaussian noise. The network architecture consists of two fully connected layers with 16 hidden units and ELU activation functions.
- **CATE Learner:** The CATE learner is modeled as an S-Learner $\mu_t(X, \tilde{U})$ and is implemented using a neural network with three fully connected layers. The first two layers have 32 hidden units, each followed by an ELU activation function. The final layer outputs a scalar, representing the estimated potential outcome.
- **MB Module:** The marginals balancing module $\tilde{g}$ is modeled as a neural network with two hidden layers, each containing 8 hidden units. ReLU activation functions are applied to the

hidden layers, and the output is constrained between $-1$ and $1$ or $0$ and $1$, using either a tanh or a sigmoid activation function, respectively.

- **PB Module:** The projections balancing module $g$ is also modeled as a neural network with two hidden layers, each containing 8 hidden units. ReLU activation functions are applied to the hidden layers, and the output is constrained between $-1$ and $1$ or $0$ and $1$, using either a tanh or a sigmoid activation function, respectively.

We use the same neural network architectures for all of our results presented in the Experiments Section 4.

**The Algorithm.** We present the full pseudo-code for MB+PB in Algorithm 1. The code consists of the training loop of the proposed model and the loss functions computation.

**Hyperparameters.** For the regularization parameter $\alpha$ is set dynamically, following the heuristic described below. We initially start with a small value for $\alpha$, and as the observed factual loss optimization stabilizes, we gradually increase the importance of the regularization term. In all of our experiments, we train for 2000 epochs. Specifically, we set $\alpha = 0.01$ for the first 1230 epochs, then linearly increase $\alpha$ from 0.01 to 100 between epochs 1230 and 1430. From epoch 1430 to 2000, we train the model with the high regularization term $\alpha = 100$. Additionally, as described in Algorithm 1, there are multiple balancing steps involved in training the MB+PB constraint. To increase the efficiency of our training process, we begin with a small number of balancing iterations (5) when $\alpha$ is small, and increase this number to 50 as $\alpha$ becomes large. Note that we use the same training strategy across all the datasets to avoid fine-tuning the hyperparameter and to have a better assessment of the presented algorithm. For the learning rates of the different neural networks they are all set at 0.001 and we use Adam as an optimizer. Finally, for the batch sizes, we use a batch size of 256, 200, and 200 for STAR, ACTG, and NSW respectively.

**Computational Resources** The experiments in this paper are not computationally expensive to conduct and were performed on the following GPU: NVIDIA GeForce RTX 3090.

### A.4 ADDITIONAL RESULTS

Here we include additional empirical results.

#### A.4.1 SYNTHETIC EXAMPLE

We begin by presenting additional results for the synthetic experiment discussed in the main text, following the approach of Kallus et al. (2019). In Figure 10, we report the $\sqrt{\varepsilon_{\text{PEHE}}}$ as a function of training epochs. Additionally, the results for the factual loss across varying degrees of confounding are provided in Figure 11.

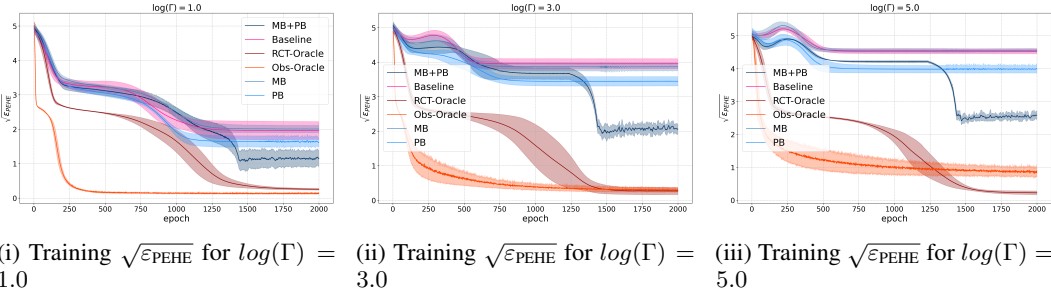

(i) Training $\sqrt{\varepsilon_{\text{PEHE}}}$ for $log(\Gamma) = 1.0$

(ii) Training $\sqrt{\varepsilon_{\text{PEHE}}}$ for $log(\Gamma) = 3.0$

(iii) Training $\sqrt{\varepsilon_{\text{PEHE}}}$ for $log(\Gamma) = 5.0$

Figure 10: Comparison of $\sqrt{\varepsilon_{\text{PEHE}}}$ across training epochs for different levels of confounding ($log(\Gamma)$).

#### A.4.2 FACTUAL LOSS COMPARISON ACROSS REAL-WORLD DATASETS

Table 2 presents a comparison of the factual loss, $\epsilon_{\text{F}}$, measured as the mean and standard deviation over 10 runs for three real-world datasets: STAR, ACTG, and NSW. We note that while the baseline

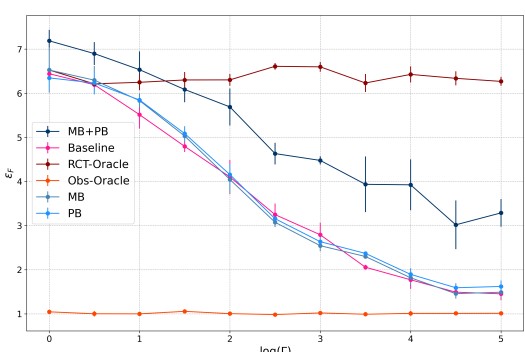

Figure 11: Factual loss comparison across different degrees of confounding.

Table 2: Comparison of the factual loss $\epsilon_F$ (Mean $\pm$ Std) across three real-world datasets. Results are presented for 10 runs.

|  | $\epsilon_F$ (Mean $\pm$ Std) | | |
|---|---|---|---|
| **Estimator** | **STAR** | **ACTG** | **NSW** |
| Baseline | $1.3 \pm 0.02$ | $1.26 \pm 0.05$ | $0.38 \pm 0.02$ |
| MB+PB (Ours) | $\mathbf{1.08} \pm 0.13$ | $\mathbf{0.72} \pm 0.03$ | $\mathbf{0.17} \pm 0.01$ |

model is designed to estimate the factual outcome, it may suffer from distributional shift as the domain of the features of the test data is different from that of the train data. Hence, learning a better causal model in that case yields better factual estimates. We conjecture that this enhanced performance is explained by the fact that our model learns a better model which makes it more robust to distributional shifts, as was formalized by (Richens & Everitt, 2024).

The baseline estimator is compared against our method, MB+PB. The results demonstrate the superiority of MB+PB in terms of lower factual loss, particularly for the STAR and NSW datasets. This reduction in factual loss indicates that our method is more effective at aligning the model predictions with the observed outcomes, thereby mitigating the effects of confounding and improving the estimation of potential outcomes.

For the STAR dataset, our method achieves a mean factual loss of $1.08 \pm 0.13$, outperforming the baseline, which has a loss of $1.3 \pm 0.02$. Similarly, the NSW dataset shows a significant improvement with MB+PB, resulting in a mean loss of $0.17 \pm 0.01$ compared to the baseline loss of $0.38 \pm 0.02$. However, for the ACTG dataset, both methods exhibit relatively close performance, with MB+PB slightly outperforming the baseline by reducing the mean loss from $1.26 \pm 0.05$ to $0.72 \pm 0.03$.

These results confirm that the MB+PB method is more robust across different datasets compared to the naive factual learner, even in terms of factual loss when there is a distributional shift, which is prevalent in real-world scenarios.

