# OpenReview forum: "Potential Outcomes Estimation Under Hidden Confounders"
_ICLR.cc/2025/Conference — Submitted to ICLR 2025_

### Official Review · Reviewer_dgih · 2024-10-16

**Soundness:** 2
**Presentation:** 2
**Contribution:** 1
**Rating:** 3
**Confidence:** 4

**Summary:**

The paper introduces two novel methods: Marginals Balancing (MB) and Projections Balancing (PB), to address the challenge of estimating Conditional Average Treatment Effects (CATE) under hidden confounding. These methods leverage outcome-only Randomized Controlled Trial (RCT) data to mitigate bias from unobserved confounders, outperforming previous approaches. The combination of MB and PB (MB+PB) further enhances performance in both synthetic and real-world datasets, providing good results even with limited RCT data.

**Strengths:**

- Empirically, the combined MB+PB method demonstrates superior performance compared to established baseline methods (albeit the experimental setting satisfies the strong assumptions imposed by the authors)
-  The idea of enforcing a balancing constraint to learn CATE is, to my knowledge, novel and interesting.

**Weaknesses:**

- **Main concern**: A critical assumption of the paper is that potential outcomes share the same distribution across both RCT and observational study. This is (almost) never the case in real-world data where even assuming the two distributions have the same conditional mean ($E_{P^{rct}}[Y|X] \equiv E_{P^{os}}[Y|X] $, aka transportability) is considered too strong. Unfortunately, this limitation is never discussed in the paper and the methodology heavily relies on this assumption.

 - **Secondary concern**: The authors assume access to only the outcomes and treatments from RCT data, without the accompanying covariates. This scenario is rare and not typically observed in practical settings, and the authors never discuss the plausibility of their setting. Clarification and examples where such a condition might realistically occur would strengthen the motivation behind the proposed methodology.


- **Concerns on writing**
  - A significant portion of the paper is dedicated to explaining that hidden confounding introduces bias in treatment effect estimation. While this is fundamentally important, it is a well-acknowledged concept in existing literature, and thus, a more concise treatment might suffice.
  - The manuscript's extensive use of bold font could be seen as distracting, reducing the emphasis intended for the most critical points.

**Questions:**

- Could the authors provide realistic scenarios where only the outcomes and treatments from an RCT are available, but not the covariates?
- Could the authors elaborate on their assumptions regarding the distribution of potential outcomes, specifically these distributions are identical for both randomized and observational data?

---

> ### Author Response · Authors · 2024-11-22
>
> Dear Reviewer dgih,
>
> Thank you for taking the time and effort to review our paper.
>
> **Assumption on Potential Outcome Distributions**
>
> You are correct that the assumption that the potential outcomes of the RCT distribution are the same as those of the observational data potential outcomes is strong and may not hold in practice. We made this assumption for ease of mathematical analysis. However, in our experimental setting, the three real-world datasets do not satisfy this assumption, and our method still demonstrates competitive performance. Please note that this is not an uncommon practice in the causal inference literature as it is the case when asserting that treatment effects are identifiable when an RCT is available. We will clarify this point in our updated manuscript.
>
> **Practical Scenarios**
>
> As noted by Reviewer hGG4, the scenario in which covariate information is missing from the RCT data is quite common. For instance, older RCTs often lack covariate data because such information was not collected at the time, e.g., some control trials were conducted before electronic health records and hence a lot of the now available covariates data is missing in the RCT experiment. Additionally, imposing a constraint that requires complete covariate information when selecting RCT candidates could introduce bias into the RCT data. Lastly, privacy concerns may lead RCT participants to withhold their covariate information.
>
> **Writing Style**
>
> The goal of Section 3 was to illustrate the impact of each regularization term on the optimization problem and its solution.
>
> Thank you for your suggestions; We will minimize the use of bold fonts.

---

> > ### Author Response · Authors · 2024-12-03
> >
> > Dear Reviewer dgih,
> >
> > As we near the end of this discussion period, we wish to extend our sincere thanks for the time and effort you have invested in reviewing our work.
> >
> > This message serves as a gentle reminder to please let us know if you have any further questions we can assist with and if you are considering adjusting your assessment of our work based on the feedback received.
> >
> > Best regards,
> >
> > The Authors

---

### Official Review · Reviewer_aUnR · 2024-10-17

**Soundness:** 2
**Presentation:** 2
**Contribution:** 2
**Rating:** 1
**Confidence:** 4

**Summary:**

The paper proposes a method for CATE estimation under unobserved confounding. It assumes the presence of a small RCT dataset to correct for the biased introduced due to the unobserved confounding.
Importantly, the paper does not reuqire the covariates of the RCT data to be observed.

**Strengths:**

- The method does not require covariate information for the RCT dataset. This is convenient in practice and differentiates the method from existing works.

**Weaknesses:**

- A discussion of methods for CATE estimation under hidden confounders (e.g. sensitivity analysis) is missing. Furthermore, a comparison to such methods is missing.
This prevents a fair assessment of the method's usefulness.
- The method is compared to a factual learner, which is not a proper baseline for comparison, as the biasedness of the learner is known and also has been discussed in the paper.
The povided empirical insights throughout Section 3 are misleading.
- The method is restricted to binary treatments. This is a disadvantage of the proposed method in comparison to existing methods for CATE estimation under hidden confounding.
- The presentation contains errors, hindering easy understanding of the line of thought. There is only limited flow in the paper.
- Limited referencing of existing works (e.g., line 70, 356)
- The paper only has limited theoretical justification.
- Evaluation: A proper evaluation of CATE estimatiors is only possible on synthetic or semi-synthetic datasets. However, the paper only considers one very low-dimensional
dataset with a very simple data generation mechanism. This is not sufficient to evaluate the general performance of the proposed method. An evaluation on further more complex
datasets with higher-dimensional unobserved confounding is necessary to properly evaluate the method.

**Questions:**

- How would the method extend to non-binary treatments?
- Line 35: In my opinion, this definition implies the consistency assumption. However, the assumption is not stated.
- Line 58: The headline of this paragraph is "performance metric". However, this is not stated in the paragraph.
- Confounding degree: Why is this introduced here? If it is considered further on, a proper introduction of sensitivity models and sensitivity analysis is necessary.
- IMHO, Section 3.1 is unnecessary and only hinders the reader's concentration. Unobserved confounding is a known topic in causal inference.
There is no need for a separate case study.
- Line 270: What is meant by pseudo-confounder?
- Line 296: Why is it reasonable to generate a pseudo-confounder? How is this done? What are the theoretical guarantees for the generated confounder?
- Proposition 3.2: What is g? Why is it necessary?
- Equation 8: This needs more mathematical explanation or at least references to works covering the respective theory
- What is the theoretical justification for the final method MB+PB?
- Figure 8: Although the combined MB+PB performs much better then the methods separately, the PEHE is still quite high (considering that the figure plots sqrt(PEHE) against log(Gamma)).
The figure thus does not aid in assessing the usefulness of the proposed method. A fair comparison with other baselines (even partial identification) would be helpful.
- Figure 8: Why does the Obs-Oracle PEHE increase with log(Gamma) even if there is no unobserved confouding introduced in the data?
- Lines 459/460: The paper states that "MB+PB shows strong robustness". However, this is not shown in the figure. Considering the scale of figure 8, MB+PB is not robust with regard to Gamma.
- Influence of RCT sample size: Again, further evaluation is necessary to draw the general conclusions stated in the paper.
- Table 1 shows that MB+PB in some cases outperforms other methods which additionally consider covariate information from the RCT. This is counter-intuitive. How can this be explained?

**Details Of Ethics Concerns:**

The paper cites false and non-existing references. This indicates unprofessional research behavior and the use and abuse of LLMs. Due to this fact, I am sadly wondering if more parts of the paper are generated by an LLM.  For more details, please see my official comment above.

---

> ### Author Response · Authors · 2024-11-13
> **Response to your wrong, unethical, careless and probably machine generated allegations/review**
>
> Reviewer aUnR,
>
> We are compelled to address several serious concerns regarding the feedback provided, which we believe are unsubstantiated, misleading and unethical. Below, we outline these issues in detail:
>
> Your serious, Unfounded and unethical Accusations:
>
> The reviewer claims that we have included fake references, generated by LLM. This claim is false and highly unprofessional. The reviewer is making baseless and highly accusatory claims and provides absolutely no evidence for it. To clarify and demonstrate the validity of our references, we have provided a complete list of all cited papers, presented in the same order as in the references section of our manuscript, along with their corresponding links:
>
> [1] Ahmed M Alaa and Mihaela Van Der Schaar. Bayesian inference of individualized treatment effects
> using multi-task gaussian processes. Advances in neural information processing systems, 30, 2017.
> Link:https://proceedings.neurips.cc/paper_files/paper/2017/file/6a508a60aa3bf9510ea6acb021c94b48-Paper.pdf
>
> [2] P Billingsley. Probability and measure. 3rd wiley. New York, 1995.
> Link: https://www.colorado.edu/amath/sites/default/files/attached-files/billingsley.pdf
>
> [3] Elise Chor, P. Lindsay Chase-Lansdale, Teresa Eckrich Sommer, Terri Sabol, Lauren Tighe, Jeanne Brooks-Gunn, Hirokazu Yoshikawa, Amanda Morris, and Christopher King. Three-year outcomes for low-income parents of young children in a two-generation education program. Journal of Research on Educational Effectiveness, 0(0):1–42, 2024
> Link: https://www.tandfonline.com/doi/abs/10.1080/19345747.2023.2273511
>
> [4] Bénédicte Colnet, Imke Mayer, Guanhua Chen, Awa Dieng, Ruohong Li, Gaël Varoquaux, JeanPhilippe Vert, Julie Josse, and Shu Yang. Causal inference methods for combining randomized trials and observational studies: a review. Statistical science, 39(1):165–191, 2024.
> Link: https://projecteuclid.org/journals/statistical-science/volume-39/issue-1/Causal-Inference-Methods-for-Combining-Randomized-Trials-and-Observational-Studies/10.1214/23-STS889.full
>
> [5] Rick Durrett. Probability: theory and examples, volume 49. Cambridge university press, 2019.
> Link: https://www.cambridge.org/core/books/probability/DD9A1907F810BB14CCFF022CDFC5677A
>
> [6] Yaxin Fang and Faming Liang. Causal-stonet: Causal inference for high-dimensional complex data. In The Twelfth International Conference on Learning Representations, 2024.
> Link: https://openreview.net/forum?id=BtZ7vCt5QY
>
> [7] Stefan Feuerriegel, Daniel Frauen, Viktoria Melnychuk, Julian Schweisthal, Katharina Hess, Alicia Curth, Sebastian Bauer, Niki Kilbertus, Isaac S Kohane, and Mihaela van der Schaar. Causal machine learning for predicting treatment outcomes. Nature Medicine, 30(4):958–968, 2024.
> Link: https://pubmed.ncbi.nlm.nih.gov/38641741/
>
> [8] Monica Gandhi, Niloufar Ameli, Peter Bacchetti, Gerald B Sharp, Audrey L French, Mary Young, Stephen J Gange, Kathryn Anastos, Susan Holman, Alexandra Levine, et al. Eligibility criteria for hiv clinical trials and generalizability of results: the gap between published reports and studyprotocols. Aids, 19(16):1885–1896, 2005.
> Link: https://pubmed.ncbi.nlm.nih.gov/16227797/
>
> [9] Thomas A Glass, Steven N Goodman, Miguel A Hernán, and Jonathan M Samet. Causal inference in public health. Annual Review of Public Health, 34:61–75, 2013. doi: 10.1146/ annurev-publhealth-031811-124606.
> Link: https://pubmed.ncbi.nlm.nih.gov/23297653/
>
>
> [10] Ruth M Greenblatt. Priority issues concerning hiv infection among women. Women’s Health Issues, 21(6):S266–S271, 2011.
> Link: https://pubmed.ncbi.nlm.nih.gov/22055678/
>
> [11] Xingzhuo Guo, Yuchen Zhang, Jianmin Wang, and Mingsheng Long. Estimating heterogeneous treatment effects: Mutual information bounds and learning algorithms. In International Conference on Machine Learning, pp. 12108–12121. PMLR, 2023.
> Link: https://proceedings.mlr.press/v202/guo23k.html
> [12] Scott M Hammer, David A Katzenstein, Michael D Hughes, Holly Gundacker, Robert T Schooley, Richard H Haubrich, W Keith Henry, Michael M Lederman, John P Phair, Manette Niu, et al. Atrial comparing nucleoside monotherapy with combination therapy in hiv-infected adults with cd4 cell counts from 200 to 500 per cubic millimeter. New England Journal of Medicine, 335(15): 1081–1090, 1996.
> Link: https://www.nejm.org/doi/full/10.1056/NEJM199610103351501
>
> [13] Tobias Hatt and Stefan Feuerriegel. Sequential deconfounding for causal inference with unobserved confounders. In Causal Learning and Reasoning, pp. 934–956. PMLR, 2024.
> Link: https://proceedings.mlr.press/v236/hatt24a.html
>
> [14] Tobias Hatt, Jeroen Berrevoets, Alicia Curth, Stefan Feuerriegel, and Mihaela van der Schaar. Combining observational and randomized data for estimating heterogeneous treatment effects. arXiv preprint arXiv:2202.12891, 2022a.
> Link: https://arxiv.org/abs/2202.12891

---

> > ### Author Response · Authors · 2024-11-13
> > **Continuation Response to your wrong, unethical, careless and probably machine generated allegations/review (1)**
> >
> > [15] Tobias Hatt, Daniel Tschernutter, and Stefan Feuerriegel. Generalizing off-policy learning under sample selection bias. In Uncertainty in Artificial Intelligence, pp. 769–779. PMLR, 2022b.
> > Link: https://proceedings.mlr.press/v180/hatt22a.html
> >
> > [16] Miguel A Hernán and James M Robins. Causal Inference: What If. Chapman & Hall/CRC, BocaRaton, 2020.
> > Link: https://www.hsph.harvard.edu/miguel-hernan/wp-content/uploads/sites/1268/2024/01/hernanrobins_WhatIf_2jan24.pdf
> >
> > [17] Jennifer L Hill. Bayesian nonparametric modeling for causal inference. Journal of Computational and Graphical Statistics, 20(1):217–240, 2011.
> > Link: https://www.tandfonline.com/doi/abs/10.1198/jcgs.2010.08162
> >
> > [18] Guido W Imbens. Causal inference in the social sciences. Annual Review of Statistics and Its Application, 11, 2024.
> > Link: https://www.annualreviews.org/content/journals/10.1146/annurev-statistics-033121-114601
> >
> > [19] Guido W Imbens and Donald B Rubin. Causal inference in statistics, social, and biomedical sciences. Cambridge University Press, 2015.
> > Link:https://www.cambridge.org/core/books/causal-inference-for-statistics-social-and-biomedical-sciences/71126BE90C58F1A431FE9B2DD07938AB
> >
> > [20] Nathan Kallus and Angela Zhou. Confounding-robust policy improvement. Advances in neural information processing systems, 31, 2018.
> > Link: https://papers.nips.cc/paper_files/paper/2018/hash/3a09a524440d44d7f19870070a5ad42f-Abstract.html
> >
> > [21] Nathan Kallus, Aahlad Manas Puli, and Uri Shalit. Removing hidden confounding by experimental grounding. Advances in neural information processing systems, 31, 2018
> > Link: https://papers.nips.cc/paper_files/paper/2018/hash/566f0ea4f6c2e947f36795c8f58ba901-Abstract.html
> >
> > [22] Nathan Kallus, Xiaojie Mao, and Angela Zhou. Interval estimation of individual-level causal effects under unobserved confounding. In The 22nd international conference on artificial intelligence and statistics, pp. 2281–2290. PMLR, 2019.
> > Link: https://proceedings.mlr.press/v89/kallus19a.html
> >
> > [23] Robert J LaLonde. Evaluating the econometric evaluations of training programs with experimental data. The American economic review, pp. 604–620, 1986.
> > Link: https://www.jstor.org/stable/1806062
> >
> > [24] Haoxuan Li, Kunhan Wu, Chunyuan Zheng, Yanghao Xiao, Hao Wang, Zhi Geng, Fuli Feng, Xiangnan He, and Peng Wu. Removing hidden confounding in recommendation: a unified multi-task learning approach. Advances in Neural Information Processing Systems, 36, 2024.
> > Link: https://openreview.net/forum?id=4IWJZjbRFj
> >
> > [25] Valentyn Melnychuk, Dennis Frauen, and Stefan Feuerriegel. Bounds on representation-induced confounding bias for treatment effect estimation. In The Twelfth International Conference on Learning Representations, 2024
> > Link: https://openreview.net/forum?id=d3xKPQVjSc.
> >
> > [26] Karl Popper. The logic of scientific discovery. Routledge, 2005.
> > Link: https://www.taylorfrancis.com/books/mono/10.4324/9780203994627/logic-scientific-discovery-karl-popper-karl-popper
> >
> > [27] Jonathan Richens and Tom Everitt. Robust agents learn causal world models. In The Twelfth International Conference on Learning Representations, 2024.
> > Link: https://openreview.net/forum?id=pOoKI3ouv1
> >
> > [28] Paul R. Rosenbaum. Observational Studies. Springer, New York, 2nd edition, 2002.
> > Link: https://link.springer.com/book/10.1007/978-1-4757-3692-2
> >
> > [29] Paul R Rosenbaum and Donald B Rubin. The central role of the propensity score in observational studies for causal effects. Biometrika, 70(1):41–55, 1983.
> > Link: https://academic.oup.com/biomet/article/70/1/41/240879
> >
> > [30] Jonas Schweisthal, Dennis Frauen, Mihaela Van Der Schaar, and Stefan Feuerriegel. Meta-learners for partially-identified treatment effects across multiple environments. In Proceedings of the 41st International Conference on Machine Learning, volume 235 of Proceedings of Machine Learning Research, pp. 43967–43985. PMLR, 21–27 Jul 2024.
> > Link: https://proceedings.mlr.press/v235/schweisthal24a.html
> >
> > [31] Uri Shalit, Fredrik D Johansson, and David Sontag. Estimating individual treatment effect: generalization bounds and algorithms. In International Conference on Machine Learning, pp. 3076–3085. PMLR, 2017.
> > Link: https://proceedings.mlr.press/v70/shalit17a.html
> >
> > [32] Claudia Shi, David Blei, and Victor Veitch. Adapting neural networks for the estimation of treatment effects. Advances in neural information processing systems, 32, 2019.
> > Link: https://papers.nips.cc/paper_files/paper/2019/hash/8fb5f8be2aa9d6c64a04e3ab9f63feee-Abstract.html
> >
> > [33] Jeffrey A Smith and Petra E Todd. Does matching overcome lalonde’s critique of nonexperimental estimators? Journal of econometrics, 125(1-2):305–353, 2005.
> > Link: https://www.sciencedirect.com/science/article/pii/S030440760400082X
> >
> > [34] Stefan Wager and Susan Athey. Estimation and inference of heterogeneous treatment effects using random forests. Journal of the American Statistical Association, 113(523):1228–1242, 2018.
> >
> > Link: https://www.tandfonline.com/doi/full/10.1080/01621459.2017.1319839

---

> > > ### Author Response · Authors · 2024-11-13
> > > **Continuation Response to your wrong, unethical, careless and probably machine generated allegations/review (2)**
> > >
> > > As shown, there is no evidence to support your claim of false references. Furthermore, suggesting that more of our manuscript may be generated by an LLM, based solely on this incorrect assertion, is speculative and lacks any concrete basis.
> > >
> > > Incorrect Line References:
> > >
> > > Several of the reviewer’s comments are linked to irrelevant line numbers, making them difficult to address:
> > > Line 58: You reference this line to be a paragraph headline, yet it points to Figure 1.
> > > Line 70: Your comment about missing references is directed at a figure caption, where no references are applicable.
> > > Line 35: There is no discussion of consistency at this line, contrary to your claim.
> > > These errors suggest a significant lack of attention to the manuscript or confusion about its content.
> > >
> > > Vague and Unhelpful Critiques:
> > > You state that our presentation "contains errors, hindering easy understanding of the line of thought," without specifying any instance of such errors. This type of feedback is too vague to be actionable, and we ask for specific examples to improve our work meaningfully.
> > >
> > > Technical Misunderstandings:
> > > You question the definition of $g$ in Proposition 3.2, although it is explicitly defined within the proposition. This indicates either a lack of familiarity with standard mathematical notations or insufficient attention to the manuscript.
> > >
> > >
> > >
> > > Conclusion:
> > >
> > > While we value constructive criticism, your review includes several baseless claims, vague critiques, and frequent inaccuracies. We believe that these issues hinder the academic integrity of the review process. We are used to low quality reviews and materially wrong reviews at ML conferences, but your baseless allegations are outrageous.
> > >
> > > We request that either you immediately show some remorse, and enter a serious apology or else we will write to technical committee with the screenshot of your dishonest comments included and request an ethics investigation in this matter.
> > >
> > >  Best regards,
> > > The Authors.

---

> > > > ### Author Response · Authors · 2024-11-13
> > > >
> > > > Dear reviewer aUnR,
> > > >
> > > > I am the student author, and I am writing to sincerely apologize for the oversight in copying author names from the NIH website, specifically from: https://pubmed.ncbi.nlm.nih.gov/38641741/
> > > >
> > > > While translating the reference from the NIH website into bibtex format, I inadvertently made errors in transcribing the full names. The original reference was provided in the following style:
> > > >
> > > > Feuerriegel S, Frauen D, Melnychuk V, Schweisthal J, Hess K, Curth A, Bauer S, Kilbertus N, Kohane IS, van der Schaar M. Causal machine learning for predicting treatment outcomes. Nat Med. 2024 Apr;30(4):958-968. doi: 10.1038/s41591-024-02902-1. Epub 2024 Apr 19. PMID: 38641741.
> > > >
> > > > In the process of converting this reference to a bibtex format, I mistakenly altered some of the names. I cannot pinpoint precisely how this happened, but I want to clarify that the reference was not generated from scratch by a language model (LLM). I used tools like Writefull and Grammarly at the sentence level (as disclosed in the submission), and these may have unintentionally modified the names.
> > > > Note that all other references are correct as I just double checked them and there were no errors in the other names.
> > > >
> > > > I want to assure you that no part of the paper was generated by an LLM, aside from sentence-level edits as disclosed.
> > > >
> > > > Best,
> > > > Student Author

---

> > > > > ### Author Response · Authors · 2024-12-03
> > > > >
> > > > > Dear Reviewer aUnR,
> > > > >
> > > > > As we near the end of this discussion period, we wish to extend our sincere thanks for the time and effort you have invested in reviewing our work.
> > > > >
> > > > > This message serves as a gentle reminder to let us know if you have any additional questions or if you could specify the lines you were referring to regarding the additional references.
> > > > >
> > > > > Below, we provide additional clarifications on some points that were not addressed in our response above:
> > > > >
> > > > > **Confounding degree:**
> > > > >
> > > > > We introduce the discussion in the background the study the sensitivity of our proposed approach to the confounding degree.
> > > > >
> > > > > **Pseudo-confounder:**
> > > > >
> > > > > The pseudo-confounder is a random variable that when introduced will make the distribution of the predicted potential outcomes the same as that of the RCT potential outcomes.
> > > > >
> > > > > **Obs-Oracle PEHE increasing:**
> > > > >
> > > > > We hypothesize that this occurs because the data size is fixed, and as confounding increases, the different treatment populations may not be adequately represented for the neural network to learn the functions optimally.
> > > > >
> > > > > **MB+PB in some cases outperforms other methods:**
> > > > >
> > > > > That is the main contribution of the paper, as we observed that competitive results can be achieved using less information compared to baseline methods.
> > > > >
> > > > > Best regards,
> > > > >
> > > > > The Authors

---

### Official Review · Reviewer_7fgj · 2024-11-04

**Soundness:** 3
**Presentation:** 3
**Contribution:** 2
**Rating:** 6
**Confidence:** 3

**Summary:**

This paper studied the problem of conditional average treatment effects (CATE) estimation. The paper proposed a new CATE estimation method in the case where in addition to a potentially large observational dataset, a small dataset from a randomized controlled trial (RCT) is also available - in particular on the outcomes are required from RCT. The proposed method is based on a pseudo-confounder generator and a CATE model which aligns the learned potential outcomes from the observational data with the outcomes observed from the RCT dataset. Numerical experiments demonstrated the effectiveness of the proposed estimation approach.

**Strengths:**

- this paper studied an important and practical problem of CATE estimation, where the estimator can be significantly biased due to the effects of unobserved confounders. The proposed method works in a setting where a large observational dataset and a small dataset from a randomized controlled trial are available - while only the outcomes in the RCT are required to be observed. I found this setting to be more realistic and applicable to many real-world cases.
- overall the paper is presented with a good clarity
- the two regularizations required by the estimation are model-agnostic, so the method is flexible to be applied to different CATE models, e.g. neural networks
- experiments with both simulation data and real-world case studies demonstrate that the proposed estimation algorithm outperform / comparable to other baselines

**Weaknesses:**

There is a lack of theoretical analysis on the two regularizations, in particular quantifying the bias reduction from each of the regularization.

**Questions:**

- How does the sample sizes of the potentially large observable data and the small RCT dataset affect the estimation outcome?
- For ACTG case study, CorNet worked better than the proposed estimator. What are the potential causes for the bad performance on this particular dataset?

---

> ### Author Response · Authors · 2024-11-22
>
> Dear Reviewer 7fgj,
>
> Thank you for taking the time and effort to review our paper.
>
> **Theoretical Analysis**
>
> We agree with the reviewer that the two proposed regularizers require further theoretical investigation from an optimization and filtering theory perspective. However, the main idea of the method is to propose these methods as potential solutions and empirically demonstrate the efficacy of the methods. Moreover, we did include a theoretical construct for the Projections Balancing regularizer where we provide an upper bound on the error in estimating the true conditional potential outcome.
>
> **Sample Sizes and Estimation Outcome**
>
> As shown in Figure 9, even a small size RCT outcomes significantly reduces the EPEHE, indicating that only a relatively small amount of RCT data is needed compared to observational data.
>
> **ACTG performance**
>
> Please note that CorNet performs better on ACTG while using more information (i.e., the *covariates* of the RCT data). In contrast, our method achieves a competitive performance while relying solely on RCT *outcomes*. We hypothesize that CorNet superior performance may result from (i) its neural net capturing the true relationship with the small RCT data and (ii) that the distributional shift between RCT and observational is not severe. These two conditions give CorNet advantages over our proposed approach.

---

> > ### Author Response · Authors · 2024-12-03
> >
> > Dear Reviewer 7fgj,
> >
> > As we near the end of this discussion period, we wish to extend our sincere thanks for the time and effort you have invested in reviewing our work.
> >
> > This message serves as a gentle reminder to please let us know if you have any further questions we can assist with and if you are considering adjusting your assessment of our work based on the feedback received.
> >
> > Best regards,
> >
> > The Authors

---

### Official Review · Reviewer_hGG4 · 2024-11-04

**Soundness:** 2
**Presentation:** 3
**Contribution:** 2
**Rating:** 3
**Confidence:** 4

**Summary:**

This paper uses RCT paired with observational data to help mitigate the impact of confounding in the observational data. The difference from other work in this literature is that it assumes no covariates are available in the RCT sample and instead tries to match the distribution of potential outcomes between the two populations.

**Strengths:**

The issue is important, and the setting where covariate information is not available for the RCT does appear in unfortunately many applications.

**Weaknesses:**

The identification strategy in the paper appears to hinge on the assumption that the potential outcome distribution is exactly the same between the two populations. This is very unlikely: RCT populations are notoriously different than observational populations in domains like health (where the privacy issues that the paper's motivation points to are most likely to arise) because trial recruitment is very far from just sampling the general population (e.g., sicker patients and patients from minority groups are typically underrepresented, along with a range of other issues) . Most work in the literature makes the weaker assumption that the CATEs are equal in the two populations and what differs is just the marginal distribution of covariates, which is much more plausible if the set of covariates available is sufficiently rich. I cannot think of any settings where imposing exact equality in the potential outcome distributions by themselves is a plausible assumption.

**Questions:**

Are there application settings where the above assumption is justified?

---

> ### Author Response · Authors · 2024-11-22
>
> Dear Reviewer hGG4,
>
> Thank you for taking the time and effort to review our paper.
>
> **Assumption on Potential Outcome Distributions**
>
> You are correct that the assumption that the potential outcomes of the RCT distribution are the same as those of the observational data potential outcomes is strong and may not hold in practice. We made this assumption for ease of mathematical analysis. However, in our experimental setting, the three real-world datasets do not satisfy this assumption, and our method still demonstrates competitive performance. Please note that this is not an uncommon practice in the causal inference literature as it is the case when asserting that treatment effects are identifiable when an RCT is available.
>
> In future work, we plan on extending the theoretical results to the more realistic scenarios, such as when a selection bias impacts the distribution of RCT participants compared to that of observational data participants.

---

> > ### Author Response · Authors · 2024-12-03
> >
> > Dear Reviewer hGG4,
> >
> > As we near the end of this discussion period, we wish to extend our sincere thanks for the time and effort you have invested in reviewing our work.
> >
> > This message serves as a gentle reminder to please let us know if you have any further questions we can assist with and if you are considering adjusting your assessment of our work based on the feedback received.
> >
> > Best regards,
> >
> > The Authors

---

### Author Response · Authors · 2024-11-13
**Demanding Investigation of Reviewer aUnR for Ethical Violation**

Dear TPC Chairs, Senior Area Chairs, Area Chairs

I am the senior author of this submission. I am writing to report a potential violation of ethics codes by reviewer aUnR.
As you can see the reviewer makes the allegation that the citations of our paper are non-existent, fake and machine generated and that the paper in part may be LLM generated.

In the response that I just posted, we gave every single citation of the paper and provided links to every citation. This demonstrates the falseness of allegations made by the reviewer.

While we value constructive criticism, the reviewer makes several baseless claims, vague and non-sensial critiques, and the review has many frequent inaccuracies. We believe that these issues hinder the academic integrity of the review process.

Note that I am a seasoned researcher and I and my students are used to low quality reviews and materially wrong reviews at ML conferences, but I find the reviewer aUnR baseless allegations very outrageous.

I am writing to you to demand an ethics investigation into reviewer aUnR  actions in this matter. From reading his/her/their review, I suspect that this review may be machine generated in part (as it is very non-coherent).

I will write to you directly (outside of the Openreview platform) to formally demand an investigation unless the reviewer aUnR retract his/her/their review.

Best regards,
Senior Author

---

> ### Comment · Reviewer_aUnR · 2024-11-13
> **Answer to unjustified accusation**
>
> Dear authors,
>
> dear PC, SAC, Ac,
>
> If there is reason to believe my review is machine-generated or in some way uncoherent, I am happy to explain my comments in more detail. Directly at the beginning of the reviewing period, I wrote a comment to the program committee about potential ethical concerns. I thought that this comment would become public during rebuttal time. Therefore, I referred to the "comment above" in my review. As this is not the case, I will repost my comment below.
>
> Of note, the reviewers make **the same mistake** in their official answer to my review, accusing me of providing unsubstantiated, misleading, and unethical reviews. The next time the authors make such a strong statement I would highly encourage them to check on the correctness of their answer.
>
> *Dear AC, SAC, PC,*
>
> *While reviewing the paper, some ethical concerns regarding truthful scientific practice and the misuse of large language models arose. I would like to ask your opinion on this matter and the potential consequences.*
>
> *Specifically, the concerns arose when reading this version of a reference stated in the submitted paper:*
>
> *Stefan Feuerriegel, **Daniel** Frauen, **Viktoria** Melnychuk, **Julian** Schweisthal, **Katharina** Hess, Alicia Curth, **Sebastian** Bauer, Niki Kilbertus, Isaac S Kohane, and Mihaela van der Schaar. Causal machine learning for predicting treatment outcomes. Nature Medicine, 30(4):958–968, 2024.*
>
> *The correct reference should read:*
>
> *Stefan Feuerriegel, **Dennis** Frauen, **Valentyn** Melnychuk, **Jonas** Schweisthal, **Konstantin** Hess, Alicia Curth, **Stefan** Bauer, Niki Kilbertus, Isaac S Kohane, and Mihaela van der Schaar. Causal machine learning for predicting treatment outcomes. Nature Medicine, 30(4):958–968, 2024.*
>
> *I can only explain this uncommon mistake through the use of LLMs. I am aware that the stated issue is not extremely severe. However, it raises questions regarding the misuse of LLMs in other parts of the work.*
>
> *Best regards,*
>
> *Reviewer aUnR*

---

> ### Author Response · Authors · 2024-11-13
> **Is this an apology?**
>
> You made very serious allegations. Now you say there were some typos.
> Since you do not apologize and do not seem to retract your statements/reviews, I will have to write to the TPC chairs and demand an investigation.
>
> Your review is non-sense, but as I said, I am used to nonsense. Your comments are preposterous.
>
> Senior author

---

> ### Author Response · Authors · 2024-11-13
> **I just had a discussion with Senior Program Chair**
>
> I just had an email discussion with the senior program chair.
>
> Again, you made some very serious allegations, and it was so serious that made me extremely upset.  I do not get upset by a paper being rejected (as I said we all know about the quality of the reviews in ML conferences), but these kinds of allegations are too much.
>
> I have foreign students who are not familiar with other national names, and one of them had a couple of typos in entering a first name or two in one reference (while working very late). I am sorry if you may be one of the authors of that paper.
>
> Just so that everyone knows, I always re-write the entire text of the paper as students' English is not perfect.  Granted that I missed a couple of typos in one of the references but the allegation that we produced this paper with LLMs is ridiculous.
>
> Now you say:
>
> "Directly at the beginning of the reviewing period, I wrote a comment to the program committee about potential ethical concerns. I thought that this comment would become public during rebuttal time. Therefore, I referred to the "comment above" in my review. As this is not the case, I will repost my comment below."
>
> What makes you think a couple of typos make a paper LLM generated while your own review is so incoherent?
>
> In anyways, my comment that your review was LLM generated was out of being provoked by your allegations. I am sorry about that. However, I maintain that your review is incoherent\ and has many issues. That said, I appreciate the time that you put into it. I do not claim to be a better reviewer than you are
>
> Best wishes
>
> Senior Author

---

> > ### Comment · Senior_Area_Chairs · 2024-11-13
> > **Untrue Claim By Senior Author**
> >
> > Hi Senior Author,
> >
> > I am your Senior Area Chair, and you did not have a discussion with me. Please refrain from claiming that you have had interactions that you have not.
> >
> > Best,
> > Your Senior Area Chair

---

> > > ### Author Response · Authors · 2024-11-13
> > > **I did have a discussion with Carl Vondrick<vondrick@cs.columbia.edu>**
> > >
> > > I had a discussion on email with Carl Vondrick which the ICLR webpage calls him a Senior Area Chair

---

> > > > ### Author Response · Authors · 2024-11-13
> > > > **My bad--his title is Senior Program Chair**
> > > >
> > > > My bad--his title is Senior Program Chair. Somehow there are too many different titles, and I thought it was the same.

---

### Meta-Review · Area_Chair_rbPH · 2024-12-23

**Metareview:**

The paper proposes to account for unobserved confounding in observational causal inference by assuming the availability of outcome data from a small RCT, which is used to train a generator model to ensure that counterfactuals from observational data match the RCT outcomes. Several empirical demonstrations show improvement of CATE estimation compared to existing techniques.

Reviewers have acknowledged the importance of the problem, particularly the fact that availability of covariate information from RCTs not being a requirement which contributes to the overall novelty of the contribution.

However, reviewers have pointed out that exactly matching counterfactuals may not be ideal given that trial populations often vary significantly compared to observational data samples, lack of sufficient theoretical analysis, lack of sufficient discussion of sensitivity analysis literature, which is highly relevant, lack of justification of when the scenario that no covariates are available from RCTs holds in practice, clarity issues in writing in that a lot of space is dedicated to explaining the bias due to hidden confounding but insufficient justification of the proposed method.

Author rebuttal has addressed some of the concerns but in general, the justification and responses are insufficient, the draft requires a considerable pass to improve clarity, provide additional justification for the setup, and justify the assumption of transportability of the CATE estimate to begin with.

Overall based on these concerns, I recommend a reject.

**Additional Comments On Reviewer Discussion:**

No additional concerns raised during discussion

---

### Decision · Program_Chairs · 2025-01-22

Reject